# Non-equilibrium anti-Stokes Raman spectroscopy for investigating Higgs modes in superconductors

Tomke E. Glier [1] ✉, Sida Tian[2], Mika Rerrer[1], Lea Westphal [1,8], Garret Lüllau [1,9], Liwen Feng[3], Jakob Dolgner[2], Rafael Haenel[2], Marta Zonno[2,4,5,10], Hiroshi Eisaki [6], Martin Greven[7], Andrea Damascelli [4,5], Stefan Kaiser [3] ✉, Dirk Manske [2] ✉ & Michael Rübhausen [1] ✉

Even before its role in electroweak symmetry breaking, the Anderson-Higgs mechanism was introduced to explain the Meissner effect in superconductors. Spontaneous symmetry-breaking yields massless phase modes representing the low-energy excitations of the Mexican-Hat potential. Only in superconductors the phase mode is shifted towards higher energies owing to the gauge field of the charged condensate. This results in a low-energy excitation spectrum governed by the Higgs mode. Consequently, the Bardeen-Cooper-Schrieffer-like Meissner effect signifies a macroscopic quantum condensate in which a photon acquires mass, representing a one-to-one analogy to high-energy physics. We report on an innovative spectroscopic technique to study symmetries and energies of the Higgs modes in the high-temperature superconductor $Bi_2Sr_2CaCu_2O_8$ after a soft quench of the Mexican-Hat potential. Population inversion induced by an initial laser pulse leads to an additional anti-Stokes Raman-scattering signal, which is consistent with polarization-dependent Higgs modes. Within Ginzburg-Landau theory, the Higgs-mode energy is connected to the Cooper-pair coherence length. Within a Bardeen-Cooper-Schrieffer weak-coupling model we develop a quantitative and coherent description of single-particle and two-particle channels. This opens the avenue for Higgs Spectroscopy in quantum condensates and provides a unique pathway to control and explore Higgs physics.

The Higgs mode is a Raman-active excitation, and was observed in $NbSe_2$ in 1980 by R. Sooryakumar and M.V. Klein[1]. The microscopic nature of this discovery was pointed out later by Y. Nambu to P. Higgs, and was seen by both as a first observation of the Higgs mode in experimental physics[2]. However, the Raman cross section of the Higgs mode in superconductors is generally very small, which results in Higgs modes that remain invisible in most Raman experiments. In $NbSe_2$, a unique interplay between a soft phonon in the charge-density

[1]Institute of Nanostructure and Solid State Physics, Universität Hamburg, Hamburg, Germany. [2]Max Planck Institute for Solid State Research, Stuttgart, Germany. [3]Institute of Solid State and Materials Physics, TUD Dresden University of Technology, Dresden, Germany. [4]Quantum Matter Institute, University of British Columbia, Vancouver, Canada. [5]Department of Physics & Astronomy, University of British Columbia, Vancouver, Canada. [6]Nanoelectronics Research Institute, National Institute of Advanced Industrial Science and Technology, Tsukuba, Ibaraki, Japan. [7]School of Physics and Astronomy, University of Minnesota, Minneapolis, USA. [8]Present address: Heinz Maier-Leibnitz Zentrum (MLZ), Technische Universität München, Garching, Germany. [9]Present address: Laboratoire MPQ, Université Paris Cité, Paris, France. [10]Present address: Synchrotron SOLEIL, Saint-Aubin, France. ✉e-mail: tomke.glier@uni-hamburg.de; stefan.kaiser@tu-dresden.de; d.manske@fkf.mpg.de; michael.ruebhausen@uni-hamburg.de

wave state and the Higgs mode in the superconducting (SC) state leads to a distinct and sharp mode slightly below $2\Delta$[3–5]. This interplay has been explored as a function of temperature and pressure[6,7], and more recently, the coupling between charge-density wave and Higgs mode has also been studied by time-resolved spectroscopy of amplitude- and phase-sensitive high harmonics[3,8]. Up to now, the observation of the Higgs mode in Raman scattering has been limited to the particular case of $NbSe_2$. In non-charge-density-wave systems, the Mexican-Hat potential must be specifically quenched, for instance by a light pulse, to make the Higgs mode observable[9].

Spontaneous Raman spectroscopy has been widely used to study excitation-energy resonances and the dynamics of the SC gap feature in high-temperature superconductors. Static Raman spectra of $Bi_2Sr_2CaCu_2O_{8+\delta}$ (Bi-2212) in $B_{1g}$ symmetry show distinct resonances between 2 eV and 3.5 eV, indicating a multicomponent origin of the excitation spectrum close to $2\Delta$[10]. Furthermore, transient time-resolved Raman scattering in the SC state enables the study of pair-breaking (PB) excitations and the dynamics of the SC order parameter[11]. By utilizing the Bose factor, time-resolved Stokes-anti-Stokes Raman scattering has been applied as a stroboscopic tool to determine transient temperatures and melting processes in highly excited states of phonons[12,13].

Advances in THz-laser technology have enabled the investigation of Higgs modes in several classes of superconductors via non-equilibrium THz spectroscopy[9,14–21]. These experiments involve either an impulsive excitation of the Higgs mode based on quench of the SC state or a drive of the Higgs mode, resulting in coherent oscillations or high-harmonic generation, respectively. Experiments on s-wave superconductors confirm that Higgs modes are stable excitations[20], whereas in d-wave superconductors they are metastable due to interactions with remanent nodal quasiparticles[19,22,23], resulting in a spectral broadening of the Higgs mode. The theoretical models are based on time-dependent Ginzburg-Landau approaches and time-dependent BCS theories in the framework of a pseudospin model[16,23–25]. Additionally, the important influence of disorder and impurity effects on high-harmonic generation was investigated from a theoretical point of view[26–28]. Furthermore, the SC state exhibits a manifold of low-energy excitations, such as pair breaking, Josephson plasmons, Bardasis-Schrieffer, or Leggett modes[22,29–31]. Some of them are difficult to distinguish from the Higgs mode in an experimental data set. Thus, it is of great importance that the Higgs modes can be classified based on the symmetry of the SC condensate, the symmetry of the quench, and the symmetry of the Higgs excitation[32]. This unique diversity of physical properties enables the exploration and control of Higgs physics in an unprecedented way and inspires the development of a transient symmetry-sensitive spectroscopic technique.

Here, we introduce a new spectroscopic technique: Non-Equilibrium Anti-Stokes Raman Scattering (NEARS). NEARS utilizes a so-called soft quench of the Mexican-Hat potential, as we will detail below, with the goal of populating Higgs modes of different symmetries, which are then probed by anti-Stokes Raman scattering. The conventional spontaneous Raman scattering signal is proportional to a four-photon Green's function where Stokes and anti-Stokes signals are generated simultaneously. Compared to conventional Raman scattering, non-equilibrium Raman scattering overpopulates the lowest-energy excitation of the superconductor, which is the metastable Higgs mode, as a consequence of the relaxation process of the free-energy landscape. This population inversion (see SI Fig. S9) can be measured by a comparison between the simultaneous Stokes and anti-Stokes signals, i.e., energy-loss and energy-gain data of the spontaneous Raman process.

## Results

Conventional Raman scattering excites quasiparticles leading to energy-loss spectroscopic features on the Stokes side (see Fig. 1a–d). In

superconductors, this technique is sensitive to low-energy excitations, such as PB excitations[33,34], see Fig. 1c, d, density-correlation functions of Josephson plasmons[31], Leggett modes[35], and Bardasis-Schrieffer modes[36,37]. Our aim is to measure the relaxation of a superconductor and the concomitant population of Higgs modes in the quasi-static limit and in nearly thermal equilibrium. As a result, excitations that are not the lowest-energy metastable states will decay more rapidly than the Higgs mode and thus remain undetectable in NEARS. This strongly supports the Higgs mode's relevance in interpreting any additional features observed in NEARS measurements.

In this work, NEARS is used to study the high-temperature superconductor Bi-2212 (optimally doped, $T_c = 92$ K, see SI Fig. S3)[38]. To verify the condition of quasi-equilibrium in the non-pumped data, we use the fact that Stokes and anti-Stokes scattering intensities in equilibrium are linked by the Bose function. We measure anti-Stokes and Stokes data and convert the Stokes spectrum to the corresponding anti-Stokes response by applying the Bose function. Matching Stokes and anti-Stokes spectra over a large low-energy range ensures that we have determined the temperature of the sample under laser illumination, excluded any background artifacts from the pulsed laser source, and avoided unwanted self-excitation effects induced by the pulsed Raman probe (see Figs. S5 and S8)[39]. Indeed, in Fig. 1b, a perfect agreement between the converted and measured anti-Stokes data is observed when assuming a heating of 9 K due to the Raman probe. We apply a scattering geometry that probes $A_{1g}$ symmetry with incidence and scattered light fields parallel to each other and rotated 45° to the $CuO_2$ plane of Bi-2212, as indicated in the pictogram of Fig. 1b. An incident photon energy of 3 eV (400 nm), a laser power of 4.8 mW, and a pulse duration of 1.2 ps were used. In the SC state, at low temperatures, and on the Stokes side, we observe the well-known phonons and excitations around twice the SC gap energy ($2\Delta \approx 60$ meV) due to the predominant PB process of Bi-2212 as expected for a sample with $T_c \approx 92$ K[10,40]. As shown in Fig. 1d, the Raman susceptibility of the PB feature is described by a tanh-function and a Lorentzian according to[10]. The PB process is sketched in Fig. 1c. At 8 K, the anti-Stokes side is essentially dark as no quasiparticles are thermally excited at T ≈ 0 K and, hence, no annihilation of excitations can occur (see 8 K data in Fig. 1b).

The NEARS experiment is illustrated in Fig. 1e. The pump quenches the Mexican-Hat potential, but allows it to relax so that the Higgs mode populates as shown in Fig. 1g. We call this scenario a soft quench, which controls the inversion population of the metastable Higgs mode via the fluence. In the normal state (Fig. 1f), with a pump (1.2 ps, 1.55 eV (800 nm)) orthogonal to the Raman probe polarizations and with a time delay of 3 ps between pump and probe, we can still apply the quasi-equilibrium approach. We can convert the measured Stokes spectra to the measured anti-Stokes data by using an additional pump heating of 3 K/mW (see Fig. S8). This is in contrast to many other experiments that use peak powers three to five orders of magnitude higher and thus explore the physics of hot electrons and hot phonons in the hard quench regime[41–44]. At 3 ps delay, fast electronic and phononic responses have already decayed[41,43,44]. Thus, following Kasha's Rule, the soft quench populates the lowest metastable excitation of the superconductor, which is the Higgs mode[45].

Indeed, in the SC state we identify an extra signal on the anti-Stokes side (see Fig. 1h) which can be assigned to the population of the Higgs modes. This feature is solely present on the anti-Stokes side, while on the pumped Stokes side we can identify a persistent suppressed PB peak, clearly indicating that the sample is still in its SC state 3 ps after the pump (see also S5 and Fig. S12). We argue in the SI (S.7) that alternative excitations such as Josephson plasmons and Bardasis-Schrieffer modes cannot be responsible for our observations. Instead, the experimental results are in agreement with Higgs excitations. Within the phenomenological Ginzburg-Landau theory, we can

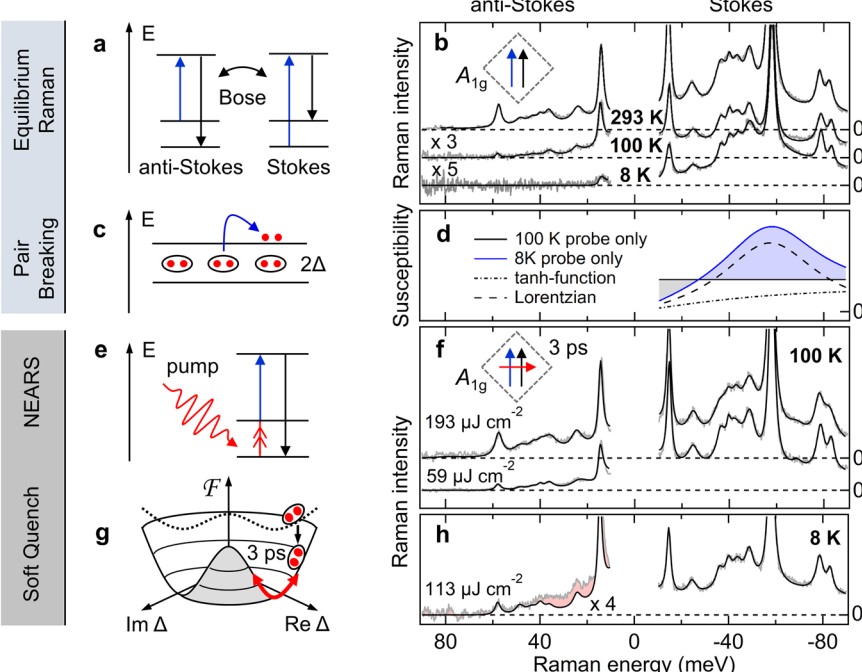

**Fig. 1 | Equilibrium vs. non-equilibrium resonant Raman scattering. a** Energy diagrams of Stokes and anti-Stokes Raman scattering. The scattering cross-sections of equilibrium Stokes (energy-loss) and anti-Stokes (energy-gain) scattering are linked by the Bose function. The Raman-probe excitation (blue) and scattered light (black) are depicted. Please note that this schematic is not drawn to scale. In all measurements, the Raman-probe excitation is 3 eV, while the Raman shift is in the meV range. **b** Equilibrium Raman data of Bi-2212 ($T_c = 92$ K[38], see Fig. S3) in an $A_{1g}$ scattering configuration at the indicated base temperatures. Measured intensities (gray) are fitted (black solid lines, see SI S.6). The dashed lines represent zero intensity for the three displayed data sets. The anti-Stokes Raman spectrum at 8 K is vanishing because the anti-Stokes Bose-function at 8 K is infinitesimally small. **c** The Raman response is dominated by PB of Cooper pairs. **d** Electronic Raman susceptibility in probe-only measurements as extracted from the parameterization (see SI S.6, Fig. S10). At 100 K the electronic background is constant (black solid line) within the presented energy range. The applied tanh function with a small $\omega_C$ approaches zero at Raman shifts below 10 meV. At 8 K, the PB feature dominates the response (blue line). The PB feature consists by a tanh-function (dash-dotted line) and a Lorentzian (dashed line). **e** After a pump (red wiggly line), the populated Higgs mode (red double arrow) can be detected by an annihilation process (anti-Stokes). **f** Pump-probe Stokes and anti-Stokes Raman spectra at a fluence of 59 μJ cm$^{-2}$ and 193 μJ cm$^{-2}$ (3ps delay, $A_{1g}$, 100 K base temperature). **g** The pump-induced modification of the free-energy landscape (dotted line) enables annihilation of Higgs oscillations (solid red line) at 3 ps delay. **h** Below $T_c$ (8 K base temperature), at an exemplary fluence of 113 μJ cm$^{-2}$ the anti-Stokes intensity cannot be described via the Stokes susceptibility and a new NEARS feature occurs (red), which we attribute to the $A_{1g}$ Higgs mode. Please note that at low temperatures, higher fluences were not applied to remain in the SC state.

connect the energy of the Higgs excitation to the Cooper-pair coherence length. Within a BCS weak-coupling model, we further develop a quantitative and coherent description of the pair-breaking excitations in the single-particle channel together with the Higgs excitations in the two-particle channel.

Figure 2 presents the fluence dependence of the Bi-2212 NEARS spectra at 8 K and at 3 ps delay in $A_{1g}$ and $B_{1g}$ symmetry, as shown in the pictograms of Fig. 2a, b. The dashed squares mark the orientation of the CuO$_2$ planes. The pump is applied along the diagonals of the CuO$_2$ planes and carries a non-zero in-plane momentum due to its incidence angle of 21.8° (see Fig. S2)[46]. In this configuration, one expects Higgs modes in both $B_{1g}$ and $A_{1g}$ probe symmetries[32]. The direct comparison between Stokes and anti-Stokes responses allows us to discriminate excitations around 2Δ from NEARS features below 2Δ. Figure 2a, b show Stokes and anti-Stokes Raman intensities for exemplary fluences between 0 μJ cm$^{-2}$ (not pumped) and 113 μJ cm$^{-2}$. Raman intensities corresponding to the parameterized PB feature on the Stokes side are depicted in blue. One can clearly see that the excitations around 2Δ in both $B_{1g}$ and $A_{1g}$ Raman probe symmetry get suppressed with increasing fluence, but remain non-zero even at the highest fluence, which demonstrates that Bi-2212 remains in the SC state (see also SI S.5 and Fig. S12). The anti-Stokes spectra are presented together with the phononic and electronic Raman intensity fitted to the Stokes side and converted to the anti-Stokes side by the Bose function. The utilized quasi-equilibrium temperatures are listed in Table S2 (see also SI S.5) and are confirmed by evaluating the superconductivity-induced features of the Stokes spectrum as a function of fluence (see Fig. S12). The additional signal, which we attribute to the Higgs mode is highlighted in red. This feature on the anti-Stokes side increases in intensity with increasing fluence. We associate this with the enhanced inversion population of the metastable Higgs mode as a consequence of the increased strength of the soft quench. In a three-level picture of population inversion (shown in Fig. S9a), an excited state is populated with a short lifetime $\tau_{relax}$[41] by quenching the Mexican-Hat. Subsequently, a metastable lower-energy state (i.e., the Higgs mode) with a longer lifetime $\tau_{Higgs} > \tau_{relax}$ is populated. Population inversion occurs if $N_{Higgs} > N_{initial}$. After exceeding a critical fluence that is required for population inversion, the anti-Stokes intensity of the metastable Higgs excitation scales with the ratio $(N_{Higgs} - N_{initial})/N_{total}$.

Figure 2c,d show the differences between the anti-Stokes data (gray in a and b) and converted Stokes fits (black solid lines in a and b), respectively. In Fig. 2e, we show the integrated Raman intensity of the Higgs modes and the PB excitation as a function of fluence. In both probe symmetries, we find an increase of the Higgs mode intensity and a concomitant decrease of the PB. The integrated intensities of the anti-Stokes Higgs difference signals as a function of fluence agree very well with the equation of population inversion (red lines) with a critical fluence of 21.7 ± 5.3 μJ cm$^{-2}$ for $A_{1g}$ and 31.6 ± 2.3 μJ cm$^{-2}$ for $B_{1g}$ symmetry (see SI S.3, eq. S24, and Fig. S9b).

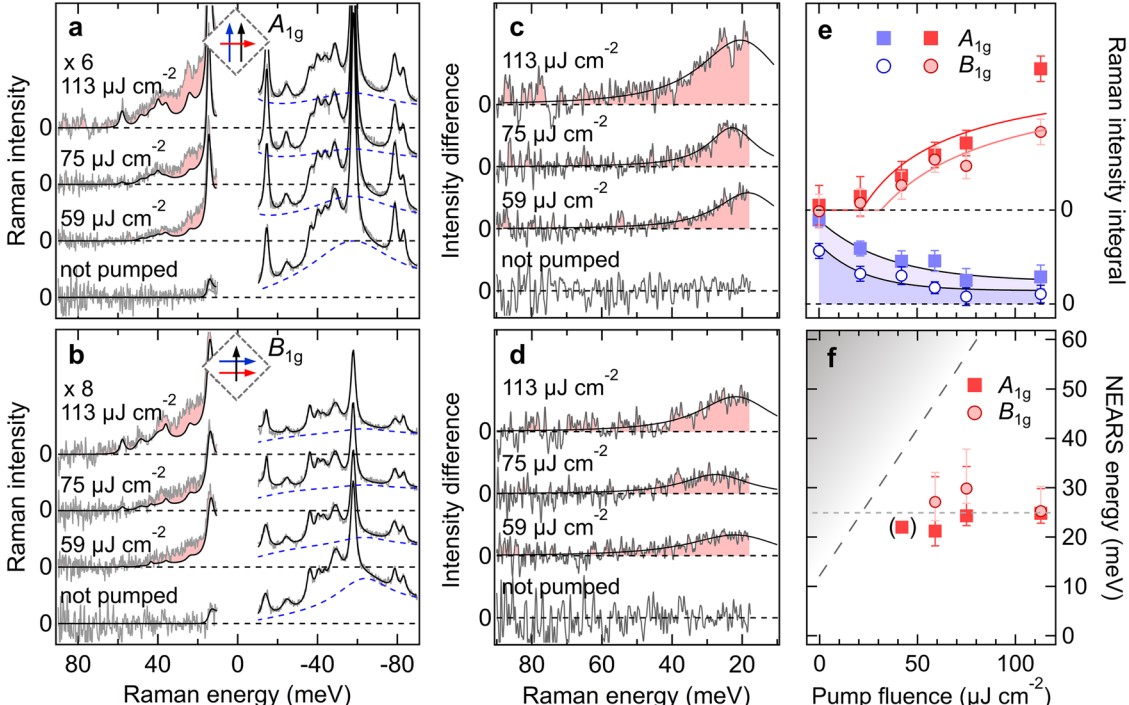

**Fig. 2 | Higgs modes as a function of fluence. a** $A_{1g}$ Pump-probe Stokes and anti-Stokes Raman spectra at 8 K base temperature and a time delay of 3 ps for selected fluences between 0 µJ cm$^{-2}$ (not pumped) and 113 µJ cm$^{-2}$ (see Fig. S6 for complete data set). Anti-Stokes intensities are scaled by a factor of 6 for visibility. Data (gray) and fits (black, see SI S.6) are shown, with dashed lines marking zero intensity. The parameterized PB response on the Stokes side is highlighted in blue. With increasing fluence, gap filling occurs, and PB amplitude decreases. Anti-Stokes spectra are shown with Stokes-side fits converted via the Bose function (black lines). Quasi-equilibrium temperatures used for the fits are listed in Table S2. At fluences larger than 50 µJ cm$^{-2}$ the in-gap electronic response is dominated by a new NEARS feature ($A_{1g}$ Higgs mode), resulting in a difference signal compared to the Stokes signal (red). **b** $B_{1g}$ Pump-probe Stokes and anti-Stokes Raman spectra at 8 K analogous to (**a**). **c** $A_{1g}$ difference signal between anti-Stokes data and Stokes-fit (corresponding to red area in a). Solid black lines represent fits following eq. (1) (see text and SI S.4). **d** $B_{1g}$ difference intensities analogous to (**c**). **e** Integrated Raman intensities of the Higgs modes (red) and of the PB response (blue) as a function of fluence. The Higgs mode intensity integrals are the integrated anti-Stokes difference spectra shown in (**c**) and (**d**). Error bars are determined based on the noise of the integrated data. Solid black lines are exponential guides to the eye. The red lines represents the equation of inversion population (see SI S.3) with a critical fluence of 21.7 ± 5.3 µJ cm$^{-2}$ for $A_{1g}$ and 31.6 ± 2.3 µJ cm$^{-2}$ for $B_{1g}$ symmetry. **f** Excitation energy of the Higgs modes vs. fluence. Error bars show the standard uncertainty of the fitted parameter $\omega_H = \sqrt{2\alpha}$ (see eq. 1). The equilibrium anti-Stokes intensity is limited at higher energies due to thermal Bose factors of the Raman intensity (gray-shaded area, see also Fig. S4) leading to asymmetric error bars.

## Discussion

A phenomenological model of the Higgs response can be obtained within Ginzburg-Landau theory utilizing a Klein-Gordon like Lagrangian. The equations of motion are derived from a Mexican-Hat potential $F(\Psi) = \alpha|\Psi|^2 + \frac{\beta}{2}|\Psi|^4$ ($\alpha < 0$) with amplitude and phase fluctuations[47]. Since the phase fluctuations are gauged out by the Anderson-Higgs mechanism, we can calculate the Green's function of the Higgs mode through the equations of motion in the optical $\mathbf{q} \rightarrow 0$ limit assuming a $\delta$ function-like quench due to a change in $\beta$ (see SI S.4). The result is a Lorentzian response $I(\omega)$

$$I(\omega) = I_0 \frac{\gamma\omega}{(\omega^2 - 2|\alpha|)^2 + (\gamma\omega)^2}, \quad (1)$$

where $\gamma$ is the phenomenological width and $2\alpha$ corresponds to the energy of the Higgs mode. The SC coherence length $\xi = \sqrt{\hbar^2/(|\alpha|4m^*)}$ is given by the Higgs-mode energy $\alpha$.

Figure 2f shows the fluence dependence of the symmetry-dependent excitation energies derived by fitting the NEARS difference data with eq. (1) as shown in Fig. 2c, d. In $A_{1g}$ and $B_{1g}$ symmetry the Higgs mode arises at around $2\alpha = 0.168 \cdot 2\Delta_0 = 10.24$ meV, corresponding to an energy of $\omega_H = \sqrt{2\alpha} = 0.41 \cdot 2\Delta_0 = 25$ meV. Using established values $m^*/m_e$ for optimally-doped cuprates of the order of $m^* = 10m_e$[48], we obtain in-plane coherence lengths of smaller than 5 nm

in agreement with other experimental observations[49]. The Higgs-mode energy is only weakly dependent on fluence as expected for a population quench.

In Fig. 3, we jointly show the PB excitations of the single-particle channel together with the Higgs modes of the two-particle channel (see also Figs. 1, 2 and Fig. S7) yielding a unified excitation landscape of the SC state in Bi-2212. These NEARS maps represent a superposition of the fluence-dependent superconductivity-induced excitations from both the Stokes and anti-Stokes spectra in $A_{1g}$ and $B_{1g}$ symmetry, respectively, identifying the $A_{1g}$ and $B_{1g}$ Higgs modes. The non-quenched energy landscape is dominated by excitations around $2\Delta$[31,33]. With increasing fluence, the $2\Delta$ excitations gradually weaken, but remain at constant energy, indicating that Bi-2212 remains in its SC state. The Higgs mode is an in-gap excitation around $0.4\ 2\Delta$ and increases in strength with fluence, showing a broadening at the highest fluence.

To interpret the NEARS data in a microscopic framework, we utilize a mean-field weak-coupling BCS theory, as discussed in detail in SI S.2. The superconductor exhibits two main contributions to its electronic Raman response in the energy range of the gap: the quasi-particle/PB response due to single-particle excitations and the collective modes' response[50]. The Higgs mode is the lowest-energy excitation in the two-particle channel and therefore plays a prominent role. The PB Raman vertices for $B_{1g}$ and $A_{1g}$ can be expressed as $\gamma_{B_{1g}} = \gamma_b \cos(2\phi)$ and $\gamma_{A_{1g}} = (1 + b_0\xi)(\gamma_0 + \gamma_1 \cos(4\phi) + \gamma_2 \cos(8\phi))$,

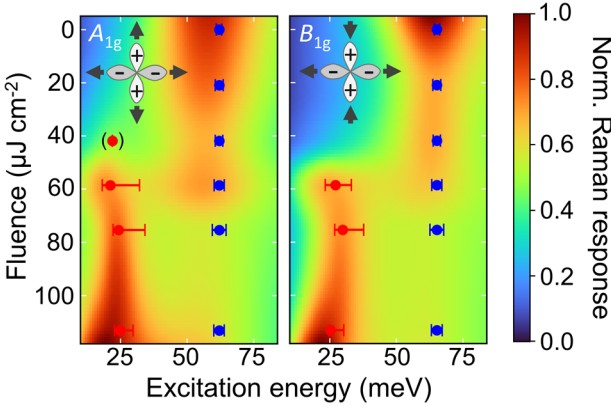

**Fig. 3 | Unified excitation spectrum of the superconductor Bi-2212.** The NEARS maps of Bi-2212 show the Raman response of the Higgs mode at around 25 meV together with the PB excitation Raman susceptibility around 60 meV in $A_{1g}$ (left) and $B_{1g}$ (right) geometry at a time delay of 3 ps. To create the NEARS maps, the fits from Fig. 2 were interpolated as a function of fluence. Raman energies of the utilized Lorentzians are shown as red and blue data points, respectively. Error bars are the standard uncertainty of the fitted frequency for the Higgs mode and PB peak, respectively. The symmetry of the Higgs modes for the respective pump/probe configuration is indicated at the top left.

respectively[34]. $\gamma_0$, $\gamma_1$, $\gamma_2$ are the coefficients in the generalized Fermi-surface harmonic expansion of $\gamma_{A_{1g}}$, and $\gamma_b$ is the coefficient in the expansion of $\gamma_{B_{1g}}$. We choose polar coordinates whose angle $\phi$ is defined as described in Fig. S14. The energy dependence of $\gamma_{A_{1g}}$ is encoded in $b_0$. The deviation from a constant density of state in the thin energy hull around the Fermi surface, inside which we assume the net-attractive electron-electron interaction, is represented by $b_1$: $N(\xi) = N_F(1 + b_1\xi)$. Please note that $b_0$ and $b_1$ break particle-hole symmetry in the Raman vertices and the density of states, accordingly. Within BCS theory, the breaking of particle-hole symmetry is required in order to obtain a finite cross-section of the Higgs mode.

In Fig. 4a, b we compare the results from our calculations with the equilibrium experimental data. In a first step, the lowest-order contribution to the $B_{1g}$ PB susceptibility is evaluated as shown in the Feynman diagram of Fig. 4a (see SI eq. S13)[34]. We obtain an excellent agreement between the parameterized, phonon-subtracted quasiparticle susceptibilities from unpumped measurements and the microscopically calculated quasiparticle response for a coefficient $\gamma_b = 0.0789 \pm 0.0002$, an electronic lifetime $\eta = 0.225 \pm 0.003$ corresponding to $2\Delta_0\eta = 13.5$ meV, and a superconducting order parameter $\Delta_0 = 30.50 \pm 0.05$ meV. Since $\eta$ and $\Delta_0$ are expected to be shared amongst all response functions, we keep $\eta$ and $\Delta_0$ constant for all further calculations.

NEARS utilizes a probe energy of 3.1 eV. It is known that the PB peak changes as a function of incident photon energy[10]. We model the Coulomb-screened $A_{1g}$ response in terms of Fermi-surface harmonics with the expansion parameters $\gamma_1$ and $\gamma_2$ (see SI S.2.4). Diagrammatically, the $A_{1g}$ PB response modified by Coulomb screening is shown in Fig. 4b (see eq. S17). Fitting $\gamma_1$ and $\gamma_2$ to the experimental phonon-subtracted $A_{1g}$ susceptibility yields $\gamma_1 = 0.0825 \pm 0.0007$, $\gamma_2 = 0.0385 \pm 0.0006$ with $\eta = 0.225 =$ const. and $\Delta_0 = 30.50$ meV = const.

The lowest-order contribution to the equilibrium Higgs response is given by the Feynman diagram shown in Fig. 4c (SI S.2.5)[4,47]. With the established parameters $\gamma_1$, $\gamma_2$, $\eta$, and $\Delta_0$ we evaluate the $A_{1g}$ response of the Higgs excitation following eq. S21 and show the resulting susceptibility together with the NEARS Higgs feature at 75 $\mu$J cm$^{-2}$ in Fig. 4c. The position of the Higgs excitation within the weak-coupling BCS model relative to $2\Delta$ depends on the symmetry of the order parameter as well as on $r = b_1/(b_0 + b_1)$[50]. Increasing the parameter $r$

shifts the peak of the Higgs response towards lower energies. The solid red line in Fig. 4c shows the calculated Higgs response for $r = 0.88$ and $\gamma_0 = 0.071$. However, Bi-2212 is a strong-coupling superconductor that goes beyond the BCS weak-coupling model. Moreover, Fig. 4c represents a non-equilibrium state that cannot be adequately captured by an equilibrium calculation. In an equilibrium calculation performed in the clean limit, we find that the Higgs susceptibility is approximately three orders of magnitude weaker than the PB peak. The NEARS experiment, on the other hand, probes the Higgs modes following population inversion in a non-equilibrium state.

Important conclusions can be drawn from our calculations. Figure 4a, b show an excellent agreement between the microscopic theory and our phonon-subtracted electronic equilibrium susceptibilities. This is remarkable, since there is no $A_{1g}$ problem[34,51] in our data taken with an incident photon energy of 3.1 eV. In addition, the Higgs mode is an in-gap excitation that can appear at energies below $2\Delta$ depending on the breaking of particle-hole symmetry.

In conclusion, by introducing NEARS, we find new in-gap excitations in a high-$T_C$ superconductor which can be assigned to the Higgs modes. This technique goes beyond conventional Raman spectroscopy for the identification of the symmetry of SC order parameters. NEARS experiments determine energy and life time enabling a first quantitative approach to the dynamics of Higgs modes in a superconductor. NEARS makes Higgs spectroscopy applicable to many materials classes characterized by the interplay of superconductivity and competing or coexisting orders[52–54]. Higgs spectroscopy depends on both the symmetry of the quench and the symmetry of the Raman polarization probing the condensate. The energy and symmetry properties of the SC Higgs field are material-dependent and enable to study and control of Higgs physics. Since the BCS-like Meissner effect requires the presence of the Higgs field, the observation of a Higgs mode can serve as a novel criterion for superconductivity.

## Methods
### Raman instrument
Spontaneous Raman measurements were performed on the UT-3 Raman spectrometer (see Fig. S1)[46]. This triple-grated spectrometer is fully achromatic due to the use of reflective optics. Excellent stray light rejection is achieved via an entrance objective with a large numerical aperture of 0.5 in a Cassegrain-type design, small focal points due to aberration-free off-axis paraboloids in combination with bilateral slits of the pre-monochromator, and an additional relay-stage equipped with two monolateral slits for setting an asymmetric bandpass for Stokes- and anti-Stokes measurements. A beam block in the entrance objective blocks reflected and emitted light in an angle of 21.8° to the vertical (see Fig. S2). In this way, a low-frequency cutoff of less than 5 cm$^{-1}$ can be achieved[46]. However, in the current experiment, the low-frequency cutoff is limited to the natural Fourier-broadening of the laser line of the pulsed laser source to approx. 80 cm$^{-1}$. We utilized a pulsed Ti:Sapphire laser system, Tsunami model HP fs 15 WP (Spectra Physics Lasers Inc., California) at a fundamental wavelength of 802 nm with a second harmonic generation (SHG) unit generating the probe wavelength of 401 nm. The pulse duration was 1.2 ± 0.1 ps monitored with an autocorrelator (APE GmbH, Berlin, Germany). The 401 nm and 802 nm beams were guided over two separate beam paths as shown in Figure S1. For time delay scans, a motorized delay line in the pump beam path was used. A $\lambda/2$ waveplate in the probe beam path allows symmetry-dependent studies. We have employed $A_{2g}+B_{1g}$ polarization by using crossed polarization and $A_{1g}+B_{2g}$ symmetry by parallel polarization between incident and scattered light with respect to the a and b axes in the CuO$_2$ planes. By rotating the sample by 45° the $A_{2g}+B_{2g}$ signal was measured for reference (see Fig. S10) showing that contributions from $A_{2g}$ and $B_{2g}$ are small. Neutral density filter units/wheels were used for fluence dependence. Figure S2 shows the details of the entrance objective consisting of four on-axis parabolic mirrors,

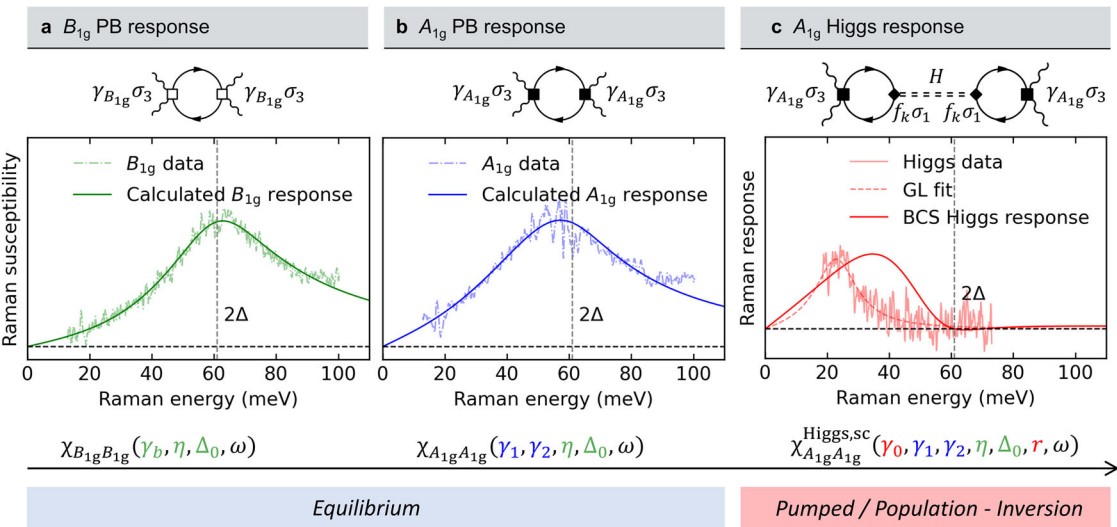

**Fig. 4 | Comparison between NEARS data and BCS model calculations. a** The experimental $B_{1g}$ electronic Raman susceptibility on the Stokes side (dash-dotted line) is extracted by parameterization and phonon subtraction as described in SI S.6. The $B_{1g}$ PB Raman response is depicted diagrammatically in the upper part of panel **a** (see also eq. S13). We find the parameters $\gamma_b = 0.0789 \pm 0.0002$, $\eta = 0.225 \pm 0.003$, and $\Delta_0 = 30.50 \pm 0.05$ meV from the $B_{1g}$ fit (green). **b** The $A_{1g}$ PB Raman response is depicted diagrammatically with filled squares representing the Coulomb-screened vertices (see also eq. S17). We keep $\eta$ and $\Delta_0$ from the $B_{1g}$ fit, and fit $\gamma_1$ and $\gamma_2$ to the $A_{1g}$ experimental data (blue, $\gamma_1 = 0.0825 \pm 0.0007$, $\gamma_2 = 0.0385 \pm 0.0006$ with $\eta = 0.225$, $\Delta_0 = 30.50$ meV.). Please note that no information of $\gamma_0$ can be obtained due to the complete screening of the isotropic $A_{1g}$

component[34]. We assume that $b_0$ and $b_1$ are small ($\Delta_0 b_0, \Delta_0 b_1 \ll 1$), therefore $A_{1g}$ has negligible dependence on $b_0$, $b_1$. **c** Finally, the $A_{1g}$ Higgs response is evaluated with the free parameters $\gamma_0$ and $r = b_1/(b_0 + b_1)$ (see Feynman diagram and eq. S21). All other parameters ($\gamma_1$, $\gamma_2$, $\eta$, $\Delta_0$) are fixed. The red solid line corresponds to $\gamma_0 = 0.071$ and $r = 0.88$. The experimental Higgs Raman response corresponds to the 75 µJ cm$^{-2}$ $A_{1g}$ data presented in Fig. 2c. The dashed line represents the fit according to eq. (1) derived within Ginzburg-Landau theory. Please note that due to the population of the Higgs mode by the soft quench, the Higgs response is a non-equilibrium response and enhanced in our experiment compared to the model. We therefore scale the intensities accordingly.

which focuses the light into the first monochromator. Pump and probe beam reach the sample at an angle of 21.8°. By this, we apply a non-zero in-plane momentum to the sample, which causes symmetry breaking and activation of the Higgs mode in $B_{1g}$ symmetry.

**Transient Stokes and anti-Stokes Raman measurements**
To characterize the beam spot size and shape and align the spatial overlap of the probe and the pump spot, a DFK 23GM021 industrial camera (The Imaging Source, Bremen, Germany) with a pixel size of 3.75 µm was positioned at the focal point of the entrance objective of the UT-3. We used a probe spot (401 nm) of FWHM = 19.5 µm x 11.1 µm (hor x ver) and a pump spot of FWHM = 22.5 µm x 16.9 µm (hor x ver). The applied probe power was 4.75 ± 0.15 mW resulting in a fluence of 35.08 ± 1.15 µJ cm$^{-2}$ at a repetition rate of 80 MHz of the Tsunami system. For the pump, fluences of 20 µJ cm$^{-2}$ to 113 µJ cm$^{-2}$ were used (see Table S2 for details). In order to establish temporal overlap of the two pulses, an ultrafast diode UPD-50-UP (Alphalas, Göttingen, Germany) was placed in the focal point of the entrance objective, where probe and pump beam were focused. A Picoscope 6402B by Pico Technology (Cambridgeshire, United Kingdom) was used to monitor the pulses and find rough temporal overlap. We then conducted reference measurements on highly oriented pyrolytic graphite (HOPG) by Alfa Aesar, Thermo Fisher Scientific (Massachusetts, USA), to calibrate the delay line. Probe-only, pump-probe, pump-only, and background measurements were taken one after the other with an integration time of 30 min each. In general, we conducted 3 repetitions per measurement to improve the signal-to-noise ratio and ensure stability over the measurement time. Stokes and anti-Stokes measurements were carried out with two different settings of the spectrometer bandpass and monolateral slits.

**Sample**
The investigated sample is an $Y_{0.08}$-substituted Bi-2212 crystal with a $T_c$ of 92 K (see Fig. S3) with the nominal composition

$Bi_{2.00}Sr_{2.00}Ca_{0.92}Y_{0.08}Cu_2O_{8+\delta}$. It is slightly underdoped as compared to $T_{c, max} = 96$ K. The sample was annealed at 500 °C in Argon. For more details on sample growth see previous work[38]. A continuous flow LHe Konti-Cryostat Spectro (CryoVac, Troisdorf, Germany) was utilized to cool the sample down to 8 K base temperature. The used cryostat is an exchange-gas cryostat with active cooling from the side of the laser impinging on the sample. At a typical optical penetration depth of 20 nm · 100 nm, this leads to an effectively base-temperature independent laser heating.

**Data treatment**
Raman spectra have been corrected for the static background detector signal and the spectral response of the spectrometer. The data was then normalized to the respective probe laser power and integration time. For NEARS data, no Bose-function correction was applied to the data itself, since the division Bose factor for anti-Stokes spectra converges to zero (see Figure S4). We, therefore, plot the Raman intensity instead of the Raman susceptibility (see Figs. 1, 2 in the main text). However, by utilizing the linkage of Stokes and anti-Stokes data via the Bose-function (see equation S29), one can calculate the anti-Stokes spectrum from the measured Stokes spectrum, and analyze the difference between the calculated anti-Stokes and the measured anti-Stokes data. As a key result of this work, we find no difference signal for all non-pumped data and pump-probe data above $T_c$ (see Fig. 1 in the main text). However, in the superconducting state and in the pump-probe measurement, we obtain a difference signal in the anti-Stokes data, which can be attributed to an overpopulation of the excited Higgs state. NEARS maps are derived from the data by plotting the superposition of the Raman susceptibility of the in-gap NEARS feature obtained from anti-Stokes data together with the Raman susceptibility of the PB feature (Stokes side) in an interpolation 2D color plot as a function of excitation energy (see also parameterization method of the PB feature in SI S.6). For this, we utilize the python class

scipy.interpolate.interp2d. In order to make this new representation of non-equilibrium Raman data more accessible to the reader, Figure S7 shows the Raman susceptibility of the NEARS feature on an energy-gain (anti-Stokes) axis, together with the superconductivity induced Raman susceptibility on the Stokes side (PB peak) on an energy-loss axis. Raman spectra (Figs. 1, 2) were fitted using the Levenberg-Marquardt fitting routine of Igor Pro (Version 6.3). To fit the electronic Raman response within our BCS model (Fig. 4), we use a Trust Region Reflective (trf) algorithm (scipy.optimize.least_squares).

### Phenomenological model and BCS theory

A phenomenological Ginzburg-Landau model can describe the charged bosonic condensate by using a Klein-Gordon like Lagrangian with a Ginzburg-Landau Mexican-hat potential $F(\Psi) = \alpha|\Psi|^2 + \frac{\beta}{2}|\Psi|^4$ ($\alpha < 0$) and for small fluctuations of the Higgs amplitude ($H$) around the ground state $|\Psi_0| = \sqrt{\frac{-\alpha}{\beta}}$ [47].

The Higgs mode, characterizing the low-energy excitation spectrum of the condensate, is Raman active and couples quadratically to the vector potential. We can calculate the equation of motion for the Higgs mode by using the Euler-Lagrange equations in the $q \to 0$ limit (see SI S.4). In order to account for the non-equilibrium experimental conditions, we can quench the order parameter within the Ginzburg-Landau theory by quenching $\alpha$, $\beta$, or both of them, since $\Psi_0$ itself depends on the ratio of $\sqrt{\frac{|\alpha|}{\beta}}$. If we quench $\alpha$ we will change the frequency of the Higgs mode to lower energy. As shown in Fig. 2f, we do not observe this behavior in the experiment. If we quench $\beta$, we will reduce the superfluid density and not change the frequency of the Higgs mode. This case fits our experimental observations. We can calculate the Green's function in the $\mathbf{q} \to 0$ case by assuming a $\delta$ function-like quench due to a change in $\beta$. See SI S.4 for more details.

We further compare the experimental data to a microscopic BCS weak-coupling theory. A weak-coupling Hamiltonian is utilized

$$\mathcal{H}(t) = \sum_{\mathbf{k},\sigma} \xi_{\mathbf{k}-\mathbf{A}(t)} c^{\dagger}_{\mathbf{k},\sigma} c_{\mathbf{k},\sigma} - \sum_{\mathbf{k},\mathbf{k}'} V_{\mathbf{k},\mathbf{k}'} c^{\dagger}_{\mathbf{k},\uparrow} c^{\dagger}_{-\mathbf{k},\downarrow} c_{-\mathbf{k}',\downarrow} c_{\mathbf{k}',\uparrow}. \quad (M1)$$

The electron dispersion $\xi_{\mathbf{k}} = \epsilon_{\mathbf{k}} - \epsilon_F$ is measured relative to the Fermi level and $c^{\dagger}_{\mathbf{k},\sigma}$ and $c_{\mathbf{k},\sigma}$ represent the electron creation or annihilation operators. A separable pairing interaction $V_{\mathbf{k},\mathbf{k}'} = V f_{\mathbf{k}} f_{\mathbf{k}'}$ with strength $V$ and symmetry function $f_k$ is used. The coupling to light is obtained by the expansion of the minimal coupling up to second order in $\mathbf{A}$

$$\xi_{\mathbf{k}-A(t)} = \xi_{\mathbf{k}} - \sum_i \partial_i \xi_{\mathbf{k}} A_i(t) + \frac{1}{2} \sum_{i,j} \partial^2_{ij} \xi_{\mathbf{k}} A_i(t) A_j(t) + \mathcal{O}(A(t)^3). \quad (M2)$$

The pair-breaking Raman vertices for $B_{1g}$ and $A_{1g}$ can be expressed as $\gamma_{B_{1g}} = \gamma_b \cos(2\phi)$ and $\gamma_{A_{1g}} = (1 + b_0\xi)(\gamma_0 + \gamma_1 \cos(4\phi) + \gamma_2 \cos(8\phi))$, respectively, as outlined in the main text[34].

The lowest order contribution to the $B_{1g}$ pair-breaking susceptibility is not Coulomb-screened[34] and can be diagrammatically represented as shown in Fig. 4a. The information about the light-matter interaction and the symmetry channel, in particular, is entirely contained in the Raman vertex function, $\gamma_{B_{1g}} \sigma_3$. With the dimensionless frequency $x = \omega/2\Delta_0 + i\eta$ we can write algebraically

$$\chi_{B_{1g}B_{1g}}(\mathbf{q} = 0, x)/N_F = \langle \gamma^2_{B_{1g}} \rangle = \gamma^2_b I_2(x), \quad (M3)$$

with the integrals $I_n(x)$ defined as

$$\langle \cos^{2(n-1)}(2\phi) \rangle = N_F \int_{-\xi_D}^{\xi_D} d\xi \int_0^{2\pi} d\phi \frac{4\Delta^2_{\mathbf{k}} \cos^{2(n-1)}(2\phi)}{E_{\mathbf{k}}(4E^2_{\mathbf{k}} - (i\omega_n)^2)}$$

$$= 2N_F \int_0^{2\pi} d\phi \frac{f^{2n}_{\mathbf{k}}/x^2}{\sqrt{f^2_{\mathbf{k}}/x^2 - 1}} \tan^{-1}\left(\frac{x}{\sqrt{f^2_{\mathbf{k}} - x^2}}\right)$$

$$= 4N_F \int_0^1 dt \frac{t^{2n}/x^2}{\sqrt{1-t^2}\sqrt{1-t^2/x^2}} \mathcal{D}(x,t) := I_n(x)$$

$$\quad (M4)$$

$$\text{with} \quad \mathcal{D}(x,t) = \left(\text{sgn}(\text{Re}(x))i\pi + \ln\left[\frac{1-\sqrt{1-t^2/x^2}}{1+\sqrt{1-t^2/x^2}}\right]\right).$$

The $A_{1g}$ response is modified by Coulomb screening of the charge fluctuations. The algebraic expression for the screened susceptibility,

$$\chi^{sc}_{A_{1g}A_{1g}} = \chi_{A_{1g}A_{1g}} - \frac{\chi_{A_{1g}\sigma_3}\chi_{\sigma_3 A_{1g}}}{\chi_{\sigma_3\sigma_3}}, \quad (M5)$$

can be expressed after Fermi-surface harmonics expansion as

$$\chi^{sc}_{A_{1g}A_{1g}} = 64\gamma^2_2 I_5 + (32\gamma_1\gamma_2 - 128\gamma^2_2)I_4 + (4\gamma^2_1 - 48\gamma_1\gamma_2 + 80\gamma^2_2)I_3$$
$$+ (-4\gamma^2_1 + 20\gamma_1\gamma_2 - 16\gamma^2_2)I_2 + (\gamma^2_1 - 2\gamma_1\gamma_2 + \gamma^2_2)I_1$$
$$- \frac{(8\gamma_2 I_3 + (2\gamma_1 - 8\gamma_2)I_2 - \gamma_1 I_1)^2}{I_1}. \quad (M6)$$

The peak position of the pair-breaking excitation in $A_{1g}$ symmetry is strongly dependent on the admixture of the higher-order Fermi-surface harmonics $\gamma_1$, $\gamma_2$.

The lowest-order contribution to the Higgs response is given by the Feynman diagram shown in Fig. 4c[4,47]. The homogeneous Higgs propagator, assuming an isotropic Fermi-surface and $T \to 0$ can be expressed as

$$H^{-1}(i\omega_n, \mathbf{q} = 0) = \sum_{\mathbf{k}} \frac{f^2_{\mathbf{k}}(4\Delta^2_{\mathbf{k}} - i\omega_n^2)}{E_{\mathbf{k}}(4E^2_{\mathbf{k}} - i\omega_n^2)} \tanh(\beta E_{\mathbf{k}}/2). \quad (M7)$$

With the established parameters $\gamma_1$, $\gamma_2$, $\eta$, and $\Delta_0$ we can evaluate the $A_{1g}$ response of the Higgs excitation following eq. S21. Details of our calculations can be found in the SI (S.2).

## Data availability

The NEARS data generated in this study have been deposited in the UHH repository (https://www.fdr.uni-hamburg.de/), see ref. 55.

## Code availability

Code is available at codeocean.com[56].

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

## Acknowledgments

The authors thank Lara Benfatto (Università di Roma "La Sapienza"), Roberto Merlin (University of Michigan), Peter Abbamonte and Lance Cooper (University of Illinois Urbana-Champaign), Kenneth Burch (Boston College), Peter Littlewood (The University of Chicago), Jim Freericks (Georgetown University), and Herbert Fotso (University at Buffalo) for inspiring and productive discussions and input. We acknowledge funding from: • U.S. Department of Energy through the University of Minnesota Center for Quantum Materials, Grant No. DE-SC-0016371, MG • Bundesministerium für Bildung und Forschung, 05K19GU5 and 05K22GU2, MRü • Max Planck-UBC-UTokyo Center for Quantum Materials, AD • Canada First Research Excellence Fund, AD • Quantum Materials and Future Technologies Program, AD • Natural Sciences and Engineering Research Council of Canada (NSERC), AD • Canada Foundation for Innovation (CFI), AD • Department of National Defense (DND), AD • British Columbia Knowledge Development Fund (BCKDF), AD • Canada Research Chairs Program, AD • CIFAR Quantum Materials Program, AD • Deutsche Forschungsgemeinschaft (DFG), SFB 1143 (project id 247310070), SK • The Würzburg-Dresden Cluster of Excellence on Complexity and Topology in Quantum Matter - ct.qmat, EXC 2147 (project id 390858490), SK • Funding by the European Union (ERC, T-Higgs, GA 101044657), SK, (Views and opinions expressed are however those of the author(s) only and do not necessarily reflect those of the European Union or the European Research Council Executive Agency. Neither the European Union nor the granting authority can be held responsible for them). We acknowledge financial support from the Open Access Publication Fund of Universität Hamburg, TG/MRü.

## Author contributions

Conceptualization: M.Rü., D.M., S.K., T.E.G.; Methodology: D.M., S.T., J.D., R.H., T.E.G., S.K., M.Rü.; Investigation: T.E.G., L.W., L.F., M.Rü., G.L., M.Re.; Resources: A.D., M.G., M.Z., H.E.; Visualization: T.E.G., M.Rü., D.M., S.K.; Formal analysis: T.E.G., M.Re.; Funding acquisition: M.Rü., D.M., S.K., A.D., M.G.; Writing - original draft: T.E.G., S.K., D.M., M.Rü., J.D., S.T.; Writing - review & editing: T.E.G., S.K., D.M., M.Rü., A.D., M.G., M.Re., J.D., S.T.

## Funding

## Competing interests

The authors declare no competing interests.
