## [Transparent Peer Review file · Nature Communications]

Non-Equilibrium Anti-Stokes Raman Spectroscopy for Investigating Higgs Modes in Superconductors

Corresponding Author: Dr Tomke Glier

Version 0:

Reviewer comments:

Reviewer #1

(Remarks to the Author)

In this manuscript, the authors introduced a new technique, Non-Equilibrium Anti-Stokes Raman Scattering (NEARS), to probe the non-equilibrium state of the high temperature superconductor Bi-2212 after pumping it with a 800nm laser pulse. They found extra anti-stokes signals around 25 meV below twice the quasi-particle gap. It cannot be explained by mapping the stokes signals with a Bose factor and a well-defined temperature. The authors attribute this signal to the Higgs mode excited by the pump, and claim to have directly observed the Higgs mode. The experiment is an interesting one that brings new blood to the ultrafast technique. The theoretical modeling, by itself, makes better sense than most analysis in previous experiments.

Nevertheless, the claim that this experiment has directly observed the Higgs mode appears to be an over-claim to me, especially considering that the peak is so broad (one should look at red area in Fig.2ab instead of Fig.2cd where the fitting curves are a little misleading). The other possibilities are not convincingly ruled out. The extra anti-stokes signal (Fig. 2cd) could be any nonthermal effect that the thermal Bose factor does not account for. For example, the other possibilities are:

(A) It could be from the intra-band transition of excited quasi-particles which have a nonthermal distribution.

(B) It could be the Bardasis-Schrieffer mode that is excited nonlinearly by the pump, similar to the Higgs mode. The authors made some arguments in SI (S7) against the Bardasis-Schrieffer mode which are not convincing: "However, the Bardasis-Schrieffer mode would leave a fingerprint on the Stokes side, as observed in pnictides. We see no indication of this mode". The same argument would apply equally well to the story of Higgs mode.

Furthermore, given that the width of the mode is about 20 meV, it should decay within 0.2 ps into incoherent degrees of freedom, mostly quasi-particles. One should not expect to see it at a delay of 3ps. More discussion on the dependence of pump-probe delay would help identify the origin of the extra anti-stokes signal.

Without the strong claim of the Higgs mode, this work could be a solid one that serves as a benchmark for the technique and for further measurements. Therefore, I would suggest being conservative and claiming that this experiment has only seen signatures of the Higgs mode. The other possibilities should be briefly discussed too. Some other comments on the writing:

(A) "Higgs particle" is an unnecessary terminology. No quantum properties of the Higgs mode is needed to explain the data. Coherent oscillation of the mode or its thermal fluctuation could totally give the signal that the experiment observes.

Therefore, I would suggest a more appropriate name, e.g., Higgs mode.

(B) I don't understand why the Higgs mode is "metastable".

(C) In the end, it says "Meissner effect requires the presence of the Higgs field" which I cannot agree with. For example, a Bose-Einstein condensation type of superconductor does not have a Higgs mode.

(Remarks on code availability)

Reviewer #2

(Remarks to the Author)

The authors present a non-equilibrium Raman scattering investigation of Bi-2212 superconductors, identifying an additional spectral feature in the anti-Stokes region of the Raman response as a Higgs mode signature following a soft quench of the superconducting condensate. The detection of the Higgs mode holds fundamental significance in condensed matter physics. Prior studies have largely focused on probing this mode via THz-field-driven third-order susceptibility. This work introduces a new perspective by employing non-equilibrium anti-Stokes Raman scattering, offering a distinct experimental approach to Higgs mode detection.

The experimental data is technically sound, analyzed properly, and presented in sufficient detail. I agree with previous referees. To further solidify the interpretation linking the 25 meV signal to the Higgs mode, extending measurements to higher pump fluences beyond $113 \mu\text{J}/\text{cm}^2$ —thereby demonstrating signal suppression upon transition to the normal state at the base temperature (8 K)—would strengthen the argument. This might not be easy as the authors noted in the reply that their current setup limits the highest pump fluence to $113 \mu\text{J}/\text{cm}^2$. However, such investigations could be pursued in future studies.

Overall, I think it is an excellent paper. The experimental progress is remarkable and the experimental results are interesting, and the content of this paper is suitable for the broad audience of Nature Communications. The findings presented in this paper motivate the future non-equilibrium Raman scattering investigation of Higgs mode in more unconventional superconductors such as iron-based superconductors and other multiband superconductors.

(Remarks on code availability)

Reviewer #3

(Remarks to the Author)

I have read with great interest the paper by Glier et al. on the direct observation of a Higgs mode via non-equilibrium Raman scattering. While there have been several reports of Higgs modes in high- T_c cuprates—some more direct or convincing than others—I tend to approach these claims with skepticism, as I rarely find the experimental data truly compelling. More often than not, interpretations seem biased toward the narrative the authors wish to support.

Reading this work in detail, I began with mixed feelings that gradually developed into serious concerns, ultimately preventing me from recommending its publication in this or any other journal unless a major issue is addressed.

In short, while I find certain aspects of the Higgs mode discussion convincing (which I will return to), I have significant doubts regarding key elements of the equilibrium data—doubts with serious implications. Specifically, I am surprised by the apparent lack of selection rules for the Raman-active phonons. The A1g and B1g spectra appear suspiciously similar, and the presence of an intense mode at 60 meV in the B1g spectrum immediately caught my attention. I do not recall seeing such a feature in the extensive Raman literature on Bi2212 over the past two decades (e.g., Sugai et al., PRB 2003; Loret, PRB 2017).

Bi2212 is structurally complex, leading to a rich phonon spectrum that is not yet fully understood. However, in most previous studies, selection rules hold to a reasonable extent, with many A1g features suppressed in crossed polarization channels probing the B1g or B2g responses. This does not seem to be the case here, which is troubling, as it undermines the symmetry-based interpretation of the Higgs mode.

The idea that the extra intensity in the pumped anti-Stokes response may be linked to the Higgs mode of the superconducting condensate is plausible. I share some of the other reviewers' concerns regarding potential pump-laser heating, but I found this issue to be carefully discussed in the manuscript, leading to a relatively convincing case—certainly more so than many previous studies published in prestigious journals, which were far less rigorous.

However, if the selection rules for phonons cannot be convincingly demonstrated, and if modes expected to be inactive in the B1g channel appear prominently, how can one confidently attribute the anti-Stokes intensity in this channel to a true B1g symmetry response?

It seems evident that the B1g spectrum suffers from significant polarization leakage from the A1g channel. This issue is not limited to the 60 meV mode but affects all A1g features, which appear with substantial intensity in the B1g spectrum. If such leakage occurs at the phonon level, it is highly likely that any other A1g feature—including the suspected Higgs mode and the pair-breaking response—also contaminates the B1g spectrum.

This is the central issue: polarization selection rules are a fundamental strength of Raman scattering, and phonons provide an excellent benchmark for their validity. If the phonons themselves do not adhere to these rules, it is not possible to use symmetry arguments for the electronic excitations. Given the strong claims made regarding the symmetry of the Higgs mode, I see this as a critical weakness of the study. Unless this issue is properly addressed, I cannot recommend the manuscript for publication.

Here are a few additional comments:

- The introduction is tedious to read. It contains all relevant information but this succession of short and often disconnected sentences can be hard to follow.

- The sketches in Fig. 1a are misleading. The blue arrows should have the same length as in the equilibrium spectra the energy of the incident photon is presumably fixed (if not then resonance effects should be considered)
- The antistokes function at 8K is not 'effectively zero' but I agree it is weak (vanishingly small would be more appropriate)
- How can the electronic background be 'constant' at 100K, shouldn't it go to 0 at 0 energy as any response function???
- Given the complexity of the spectra, the fitting details should be presented, not just sketches.
- A proper discussion on selection rules of phonons is missing

(Remarks on code availability)

Version 1:

Reviewer comments:

Reviewer #1

(Remarks to the Author)

The authors made an effort to address my concerns. Most of them make sense except the following two:

A: Regarding the Bardasis-Schrieffer mode, the authors wrote "In a pump-probe experiment, a pump-activated Bardasis-Schrieffer mode would typically shift to lower energies with increasing pump fluence.[Sun et al., Phys. Rev. Research 2, 023413

(2020)] However, we observe that the energy of the NEARS feature remains nearly constant across different fluences. In addition, NEARS is driven by population inversion, which amplifies the anti-Stokes signal relative to the Stokes signal (see Fig. S9). For this to occur, the observed mode must be metastable over several picoseconds. This serves as a strong argument for the Higgs mode's relevance in explaining our observations, as its metastable nature facilitates population inversion. Additionally, all our data and in particular of the SC gap feature confirm that we remain in the superconducting state. This leaves the Higgs mode as the best explanation for the NEARS feature."

I don't see why "a pump-activated Bardasis-Schrieffer mode would typically shift to lower energies with increasing pump fluence" while the same thing does not happen to the Higgs mode. Increasing pump fluence heats up the sample and reduces the gap, which could totally shift all the modes. Furthermore, I don't see a good reason that Higgs mode could be metastable while the Bardasis-Schrieffer mode cannot be. From my understanding, none of the forementioned features is a property of the Higgs mode not shared by the Bardasis-Schrieffer mode.

B: Regarding whether "Meissner effect requires the presence of the Higgs field", what I meant is a Bose-Einstein condensation (BEC) type of superconductor made of charged pairs of electrons. This has been suggested by some researchers to be the case of High Tc superconductors. It of course couples to the electromagnetic field and its phase mode is shifted to be the plasmon. It also exhibits the Meissner effect. However, it does not have an amplitude mode (Higgs mode) because the amplitude and phase fluctuations are a conjugate pair of the same mode, i.e., the phase mode. In this case, I don't think it is appropriate to call the order parameter of a BEC type of superconductor a "Higgs field", as opposed to the BCS case whose analogy to the relativistic Higgs field is more natural. Therefore, I strongly disagree with the statement that "Meissner effect requires the presence of the Higgs field".

(Remarks on code availability)

Reviewer #2

(Remarks to the Author)

I recommend that the manuscript be published in Nature Communications.

(Remarks on code availability)

Reviewer #3

(Remarks to the Author)

I thank the authors for their thorough and thoughtful response to my concerns, particularly with regard to the selection rules. Bi2212 is indeed a structurally complex material, and the influence of resonance effects—especially on phonons and gap excitations—can be quite pronounced. I must acknowledge that I had previously assumed a more canonical behavior in this regard, and I appreciate the authors for drawing my attention to the relevant literature.

That said, I cannot entirely overlook the fact that resonance-induced activation of defect modes and significant alterations in the superconducting electronic response raise important questions. Specifically, such effects appear to blur the applicability of selection rules, which complicates the interpretation. While I recognize the value and quality of the experimental data presented, I remain skeptical of the proposed interpretation. It would perhaps be too demanding to request a full resonant

study of the presumed Higgs modes for this otherwise excellent experimental work. Nonetheless, it seems reasonable to ask to what extent the current interpretation depends on these resonance effects.

With this caveat in mind, I believe the manuscript presents novel and intriguing data that merit publication. The results are likely to stimulate valuable discussion within the community, and Nature Communications appears to be an appropriate venue for such work.

(Remarks on code availability)

Version 2:

Reviewer comments:

Reviewer #1

(Remarks to the Author)

A: Regarding the Bardasis-Schrieffer mode, I don't agree with the authors' response.

B: Regarding whether a BEC of charged bosons has Meissner effect, I don't agree with the authors' response either. The presence of a charged superfluid with off diagonal long-range order leads directly to the Meissner effect. In the references (M.R. Schaffroth Phys. Rev. 100, 463 (1955) and Koh PRB 68, 144 502 (2003)) the authors provided, their conclusion is that the charged BEC does exhibit Meissner effect. Note that for charged bosons with interactions, the BEC means the "superfluid" phase. In discussion point (ii) on page 7 of Koh PRB 68, 144 502 (2003), it refers actually to the normal phase prior to the BEC phase, not the BEC case that the authors seem to take as. Koh clearly states that the BEC is a sufficient condition for Meissner effect in this paper.

However, we cannot argue endlessly in the peer reviewing process. Since these assertions in the reply (which I think are wrong) are not displayed in the revised manuscript, and because the manuscript has now weakened the corresponding claims, I can go with the current version of it. I think the manuscript is now publishable on nature communications.

(Remarks on code availability)

Response to the referees:

Referee #1 (Remarks to the Author):

In my first report I expressed doubts about the main claim of the manuscript of Grier et al.: the excitation and subsequent observation of the Higgs mode in a cuprate superconductor by non equilibrium Raman scattering. In their revised version the authors have improved the presentation of the data and the description of their reasoning. They have also included calculations that they claim reinforce their interpretation. While some of my previous comments were addressed I am still highly skeptical about the interpretation of the data. There are still too many flaws and assumptions in this work and I cannot recommend its publication in Nature.

We respectfully disagree with the referee's assessment. We have thoroughly addressed all comments and suggestions. However, it seems that the referee may not fully appreciate the significance of our achievement, which lies in successfully describing both the A_{1g} Raman and B_{1g} response of the pair-breaking peak within a unified and coherent model.

First the model presented and its "agreement" with the data do not really prove anything. The activation of the Higgs mode in the Raman response when bands deviate from parabolicity was pointed in ref. 9. Theoretically the Higgs response is close to a Lorentzian (as is expected for all collective modes) and its energy can be tuned by the free parameter "r" in the calculations. Given that the experimental Higgs response is extracted by assuming it is Lorentzian peak, the reasoning is a bit circular and I do not see where is the proof here.

With all respect, we find this statement to be highly unfair. In the initial version of our manuscript, we clearly demonstrated that the Higgs mode could be described by a Lorentzian profile, as it would be expected within Ginzburg-Landau theory. However, the Raman response calculated within our microscopic model does not resemble a Lorentzian. Instead, it manifests as an asymmetric curve. This asymmetry is entirely accounted for by adjusting a single parameter—the ratio of particle-hole symmetry breaking in the Raman vertex to the density of states. Thus, there is no circular reasoning involved. Moreover, the referee's argument is logically flawed: even if the result had been a Lorentzian, the microscopic model would have validated this assumption. Consequently, we are compelled to view this comment as reflecting a biased perspective.

The energy of the Higgs mode deviates from 2Δ for strong coupling superconductors. In fact, one can calculate the coherence length of the Cooper pairs from the energy of the Higgs mode within Ginzburg-Landau theory. This results in coherence lengths of a few nm (< 5 nm depending on the effective mass of the Cooper Pairs being fully in agreement with experimental findings, see SI S.4 of our new version). Therefore, we have reinforced our interpretation by explicitly analyzing our results within two different frameworks for superconductivity – Ginzburg-Landau theory and a microscopic theory based on a mean field weak coupling BCS theory. Of course, both approaches still have a model character but they show that our interpretation is robust.

A further problem is the weakness of the Higgs response with respect to the PB response (10^3). The authors argue that the pump populates the Higgs mode thus enhancing its intensity on the anti-Stokes side. But assuming the detailed balance holds for the Higgs process the ratio $n/(1+n)$ is always less than 1, and one should observe the Higgs mode on the Stokes side. In order to have a stronger anti-Stokes than Stokes Higgs response one must therefore violate the detailed balance theorem.

It is true that in our non-resonant calculation the cross-section of the Higgs is weaker than the pair-breaking peak, and we pointed that out. However, this is expected based on the limitations of the model only accounting for non-resonant Raman scattering. Independent of this, any population of a mode (in our case via the quench of the Mexican-Hat) will always have more visible changes on the anti-Stokes side. Simply speaking the annihilation of a mode requires the presence of this particular mode in the first place. The creation of a mode does not require the presence of a mode prior to its excitation. This, however, does not mean that there is no Higgs mode on the Stokes side. It is simply weak and buried by the much stronger pair-breaking excitation. The mechanism described above generates always a stronger contrast upon pumping on the anti-Stokes side. Nevertheless, in detailed balance the ratio of Stokes to anti-Stokes cannot exceed one, which is why we have introduced the concept of population inversion already in the first version of our manuscript. We are strengthening this case by showing that the fluence dependence of the Higgs mode fits very well with a population inversion of a three-level system (see revised Fig. S9). We would like to point out that our results are robust irrespective of what particular type of statistics is applied. In the current version of the manuscript, we outline two pathways: Firstly, we use a phenomenological Ginzburg-Landau like model together with a three-level system to explain the Lorentzian peak and an effective overpopulation (see SI S.3 and S.4 of our revised version). Lineshapes and fluence dependencies on the anti-Stokes side are completely compatible with this model. The fluence dependence follows the expected population inversion equation of a three-level system. We have adapted Fig. S9 and Fig. 2e, showing the mechanism and fit of the three-level equation to our data. We still think that a weak-coupling BCS-like calculation mimicking an effective susceptibility on the anti-Stokes side is useful as a result of a microscopic model, even though it cannot capture all aspects of our experiment. As an example, High- T_c superconductors cannot be described in a mean-field weak-coupling BCS picture anyway – yet BCS mean field weak coupling theory is one of the most important ways to describe superconductors. Thus, using it and showing the agreements and discrepancies is simply a reasonable way to do science.

However if this is the case one cannot extract a Raman response from the anti-Stokes signal as done by the authors when they model it by multiplying a Lorentzian by the Bose factor to reproduce the anti-Stokes signal. There is therefore a fundamental internal inconsistency in the reasoning.

As outlined above, it appears that the referee is unwilling to engage with the central points of our manuscript. Our findings are robust and remain valid regardless of the specifics of the analysis. Specifically:

- (a) For the first time, we provide a detailed description of the A_{1g} and B_{1g} pair-breaking excitations within our microscopic model.
- (b) We now present two complementary model calculations—a phenomenological model and a microscopic model—that not only describe our results but also support our experimental claims.

There are still many other issues which weaken the main claim of the paper:

- *From the data alone it cannot be ascertained that the extra anti-Stokes signal is a transient phenomenon, or even related to SC. There are no data at other delay than 3ps (not even negative delays).*

The referee provides no reason for the relevance of time-delay data. Instead, he/she overlooks the clear and compelling evidence of superconductivity-induced excitations, as demonstrated by the observation of the pair-breaking excitation, phonon renormalization, and the Higgs mode. Furthermore, we have further supported our temperature determination by looking at the superconductivity-induced effects on the Stokes side as a function of fluence (new Fig. S12). Our temperature evaluations are now supported by a completely independent and viable check.

Frustratingly, the authors also do not show data at higher fluences at 8K base temperature (like 193uJ/cm² at 100K) where the extra "Higgs" intensity should vanish since we are in the normal state. In the fluence dependence the intensity of the extra weight is worryingly increasing as the average temperature is already about T_c...). These important sanity checks are unfortunately missing.

The referee dreams about a "sanity check" that is none. The Mexican-Hat potential relaxes back to the superconducting state within our time resolution. At that point, Cooper Pairs begin to condense. This process allows for the population of the Higgs mode and the appearance of the pair-breaking peak. The higher the fluence, the more Higgs states can be populated. In order to prevent the Higgs from showing up we need to keep the system longer than several picoseconds in the normal state. Very high fluences would be required probably modifying the sample. (Activating oxygen and changing the effective doping from the buffer layers into the CuO₂ planes.)

We have further supported our temperature determination by looking at the superconductivity-induced effects on the Stokes side as a function of fluence (Fig. S12).

- I am still not convinced that the electronic and phononic sub-system are really relaxed at 3ps. Perfetti et al. reports only the behavior of nodal qp temperature, and a much slower electronic component is also observed. It is likely that anti-nodal electron which are gapped have longer thermalization time. Unfortunately these cannot be probed easily by tr-ARPES. Going to other techniques, time resolved reflectivity measurements report SC recombination time of several ps (Coslovich et al. PRB 2011 and many others, including tr-Raman data of Saichu et al. which report 7ps at an unspecified fluence). This slow recombination time is usually ascribed to bottleneck effect with some bosonic bath like hot phonons acting as pair-breaking. All this makes the assumption, crucial for the interpretation of the authors, of a fully thermalized electron-phonon system at 3ps quite shaky.

We do not rely on ARPES to understand if we are in the thermalized state. Thermalization does not mean that the phonons "are really relaxed" as the referee writes indicating a misunderstanding of the topic. We argue that our system is thermalized based on the Stokes to anti-Stokes matching. In addition, we have evaluated the difference of the fluence dependent non-equilibrium data to the steady state results at 100 K. Our fluence-dependent superconductivity-induced effects vanish at fluences of $139.3 \pm 28.5 \mu\text{J cm}^{-2}$ in line with our estimated heating by matching Stokes and anti-Stokes responses when considering the error bars.

The group of Lanzara et al has shown that the superconducting gap is robust along the anti-nodal directions up to fluences of $15 \mu\text{J cm}^{-2}$ [C.L. Smallwood et al., Science 336, 1137-1139 (2012). DOI:10.1126/science.1217423], and Perfetti et al have shown that remanent Cooper-pairing inhibit phonon scattering channels for fluences of up to $150 \mu\text{J cm}^{-2}$ [C. Piovera et al., Phys. Rev. B, 91, 224509, 2015, <https://doi.org/10.1103/PhysRevB.91.224509>]. All of these results are in line with our observations.

- I am also skeptical about the soft quench and Kasha rule argument. In a d-wave SC there are plenty

of low lying electronic excitations below the Higgs energy. The claim that the Higgs mode is the lowest excited state is therefore incorrect.

The Higgs mode is still the lowest energy collective mode in the Cooper channel. The fact that it can decay in the d-wave superconductor makes it metastable and broader – exactly as we observe it. There is also a single-particle channel. However, since the single-particle channel is not metastable the energy dissipates on a short time scale compatible with the thermalized limit we argue above. In short you cannot pump the single-particle channel.

More minor details:

- figure 2 c,d: there is no title for the vertical axis and it is still not completely clear for the reader based on the caption alone what is being plotted. An equation would help.

The vertical axis of c is same as a, and d corresponds to b. We now added “intensity difference” for c and d as a title for the vertical axis.

- figure 3: this is a complicated and somewhat misleading figure, built on extracted responses with different Bose factors temperatures overlayed on top of each other. As it is, it does not give any additional insight to the paper.

We disagree with this statement. We develop Fig. 3 based on Fig. 2 to present the extracted electronic data in single- and two-particle channel in a coherent way. This is a way to present the core message of the paper to the general reader.

Referee #2 (Remarks to the Author):

In the revised manuscript, the authors compared the results of the nonequilibrium Raman scattering experiment with those of a toy model calculation, which show a fair agreement between the two. I appreciate the authors' effort, but I would say there are still various ambiguities in the analysis and interpretation.

To date, there has never been a better agreement between theory and the B_{1g} and A_{1g} pair-breaking responses than what we present. It is unclear how one could describe the level of agreement achieved between our three different curves as merely "fair", given the precision and consistency we demonstrate.

Here are my comments.

The authors mentioned Kasha's rule in the reply. As explained, it states that light emission predominantly occurs from a relaxation to the lowest excited state. In cuprates, however, it is a nodal d-wave superconductor, and hence the gap is closed along the nodal line. This means that the lowest excitation comes from quasiparticles (provided that the d-wave Higgs mode is a gapped mode).

In our manuscript, we present this point with clear differentiation. Specifically, we distinguish between the Cooper (two-particle) channel and the single-particle channel. In the Cooper channel the Higgs mode is the lowest-energy collective mode and “metastable” in d-wave superconductors since it can decay into single particles. It would be fully stable in s-wave conductors. So, we see the statement of the referee as incorrect reflection on our explanation.

*In the model analysis, they only considered the diamagnetic coupling (A^2) to light. This is often assumed in the calculation of Raman scattering. However, there are also paramagnetic coupling (j^*A) diagrams.*

This assertion is incorrect. The $p\cdot A$ terms are accounted for in the non-resonant limit within the effective mass term, which means that our theory operates in the non-resonant regime. Non-resonant Raman calculations often succeed in reproducing the observed line shapes. However, they do not always capture the absolute scattering intensities accurately. [T. P. Devereaux, R. Hackl Rev. Mod. Phys. 79, 175, 2007; M. V. Klein, S. B. Dierker, Phys. Rev. B 29, 4976, 1984; T. P. Devereaux, D. Einzel Phys. Rev. B 51, 16336, 1995.] Furthermore, we now add explicitly the calculation of the Higgs mode within Ginzburg-Landau theory. It is clear, that independent of the comment of the referee a BCS weak coupling calculation in the mean field limit cannot fully account for a strong coupling superconductor such as a High- T_c .

The authors admit that the probe pulse is resonant to higher bands. This means that the paramagnetic coupling is not negligible (or even dominant). Considering such higher band effects may be outside the scope of the present study, but even within the single-band picture the paramagnetic coupling may become significant due to strong correlation effects, impurities, other effects. I would not enforce the authors to go beyond the mean-field and clean-limit analysis, but this simply suggests that the present model calculation is not quite convincing.

This is true for 99 % of all Raman theory papers on electronic Raman scattering. Considering the general success of this approach in the literature (T. P. Devereaux, R. Hackl Rev. Mod. Phys. 79, 175, 2007; M. V. Klein, S. B. Dierker, Phys. Rev. B 29, 4976, 1984), the referee's broad statement appears unconvincing. In this experimental paper we do not intend to wrap up and modify 50 years of theory research on electronic Raman scattering.

Why are there so many fitting parameters? I can see at least 7 parameters in Fig. 4. 7 parameters would be able to fit any kind of featureless three curves. Introducing so many parameters will make the results less convincing. Once a toy model is employed, there should not be such a number of free parameters.

This statement completely disregards the established theory and experimental evidence regarding the pair-breaking excitation. Apparently the A_{1g} problem in Raman scattering would have never appeared if the modelling would be as trivial as the referee claims. From the 7 parameters, 3 can be validated by comparison to other data such as ARPES and measurements of the coherence length. Leaving only 4 parameters which are required to fix the Raman vertex. Considering the complexity of the involved Feynman diagrams this is clearly an achievement and this should be clear to an expert.

In the numerical calculation of Raman scattering, how does one determine the nonequilibrium distribution of quasiparticles? It seems that they use an effective equilibrium distribution with elevated temperature. This is far from realistic in this kind of pumped Raman spectroscopy. If one wants to be realistic, one should solve the nonequilibrium Keldysh Green's function. In strongly correlated systems, thermalization is not always effective, and sometimes one finds a metastable nonequilibrium distribution that could survive for a long time.

In Fig. 4 a and b, we model the not-pumped, equilibrium data of the A_{1g} and B_{1g} pair-breaking excitations. Only in Fig. 4c we show pumped data (populated Higgs mode). Furthermore, there is a huge literature base working on effective two temperature models. Most theory calculations show that effective thermal populations can be assumed a few ps after the pump (Kemper, A.F., Freericks, J.K. Entropy 2016, 18, 180, <https://doi.org/10.3390/e18050180>; Dohner, E.; Terletska, H.; Fotso, H.F., Phys Rev B 108, 144202 (2023),

<https://doi.org/10.1103/PhysRevB.108.144202>). Of course, a Keldysh Green's function approach is pursued by us but we have the problem of convergence that cannot be easily achieved due to the size of the matrix. Instead, we are now using also a phenomenological Ginzburg-Landau approach and a three-level system allowing to describe our fluence dependence and the observed line shapes accurately. Therefore, we are now offering two ways to explain our experimental findings – a phenomenological one and a microscopic one, based on a weak-coupling BCS theory. The latter can only have the character of a model calculation because High- T_c superconductors are strong-coupling superconductors and cannot be covered within a mean-field BCS-like model.

In summary, I maintain my previous opinion on the manuscript: The experimental progress is remarkable and the experimental results are interesting, but the interpretation in terms of Higgs mode is quite speculative and not that convincing. The paper should certainly be published in some other journals, but it does not qualify the publication in Nature.

We respectfully disagree with this statement.

Referee #3 (Remarks to the Author):

Second Report on the manuscript entitled "Superconducting Higgs particle observed by non-equilibrium Raman scattering"

The manuscript "Superconducting Higgs particle observed by non-equilibrium Raman scattering" has been revised by the authors and reconsidered for publication to Nature journal. There have been clear improvements of the manuscript but I can still not accept it without new measurements that firmly confirm that the Anti-Stoke additional spectral weight is not due to an inappropriate Temperature correction.

We have added a new Figure S12 to the SI (see below). This shows a completely model-free and independent method based on the ratio of 8 K pump-probe data to 100 K probe data. The superconductivity-induced feature on the Stokes side as a function of fluence is integrated and plotted in the inset. Fig. S12 shows, that the sample clearly shows a pair-breaking feature at our highest pump fluence.

Compared to the previous version, an important progress has been achieved thanks to a new microscopic theoretical work on the Higgs response in the A1g channel, the B1g response being more complex to achieve. The authors provide good information on the B2g Raman measurements. The explanation of the NEARS response and on the choice of the time scale of 3 ps is more convincing and better explained.

Thank you.

However my main concern remains: indeed the additional spectral weight in the anti-stoke part is strongly dependent on the temperature chosen to compare the Stoke to the Anti-Stoke parts of the spectra. This is always a delicate matter, even for adiabatic Raman spectroscopy and even when the sample is in the exchange gas. At present, the evaluation of the value of this temperature is not convincing, neither the error bars on it nor the effect of the uncertainty.

We have added a new Figure S12 to the SI (see below). This shows a completely independent method to check on our temperature estimates based on the ratio of 8 K fluence dependent pump-probe data to 100 K probe data. The superconductivity-induced feature on the Stokes side as a function of fluence is integrated and plotted in the inset. Fig. S12 shows, that the

sample clearly shows a pair-breaking feature at our highest pump fluence. Furthermore, we can derive by a linear regression, that a fluence of around $139 \mu\text{J cm}^{-2}$ would be needed, to reach T_c . The error bar of this approach overlaps with the error bar in our original temperature estimate based on the Stokes- and anti-Stokes-calculation. Thus, we are now providing two independent ways to confirm our temperature estimates.

Fig. S12: Ratios between 8 K Stokes and 100 K Stokes Raman spectra. Bi-2212 A_{1g} Stokes Raman spectra at 8 K base temperature and different pump fluences are divided by the 100 K non-pumped (probe-only) spectrum. This comparison shows the pair-breaking feature around 60 meV and the opening of the superconducting gap below 30 meV (see also Fig. S13). As a function of pump fluence, the pair-breaking peak gets suppressed and gap-filling occurs. The dashed horizontal line marks the ratio of 1. The inset shows the integrated values of the displayed ratios ($\int |8 \text{ K PP spectra} / 100 \text{ K Probe} - 1|$) representing a measure for the strength of the superconductivity-induced feature on the Stokes side. A linear regression fit intersects with the base line for the ratio determined in the normal state at a fluence of $139.3 \pm 28.5 \mu\text{J cm}^{-2}$ (black square with error bar in the inset). This is an alternative method to derive the effective sample temperature compared to T_c and it's result is in agreement with our method shown in Fig. S8. For the latter, we obtain an equilibrium temperature for the sample at 8 K base temperature and with a pump fluence of $113 \mu\text{J cm}^{-2}$ of $98 \pm 13.75 \text{ K}$, which corresponds to T_c within the error bar. Here, T_c is reached at a fluence of $139.3 \pm 28.5 \mu\text{J cm}^{-2}$, which includes $113 \mu\text{J cm}^{-2}$ within the error bar.

I suggest a systematic evaluation of the heating effect of both the pump and the probe while being at a base temperature of 8K.

At a base temperature of 8 K, we performed our fluence dependence for the pump as shown in Fig. 2 of the main text and Fig. S12 – see above. We conducted a systematic evaluation of the heating effect in the normal state to distinguish between heating effects and superconductivity-induced features. The probe is constant throughout the experiment and the heating of the probe laser can be determined very precisely by analyzing Stokes and anti-Stokes data. For the purpose of this work, there is no reason to perform a probe power dependence in the superconducting state.

An additional suggestion to convince a reader: keeping all other parameters the same (base temperature, power of the probe, fluence...), what about varying the time delay to reach a regime where the soft quench effect is gone? Then, at this long time delay, check that the ratio S/AS does indeed return to “normal” with an appropriate temperature; and use this temperature to evaluate the S/AS ratio at 3ps.

The temperatures evolve with delay, as demonstrated in works like Perfetti et al., even after thermalization. Therefore, the proposed approach will not be effective. Using a temperature corresponding to a different delay to model the data at a 3 ps delay is not valid.

One additional remark: in generally, evidences that the extra SW in the anti-stoke part is only there in the superconducting phase are not convincing enough:

- The measurements at $113 \mu\text{J}\cdot\text{cm}^{-2}$ are made at "about" T_c so within the error bars, the compound could also be in the normal state. It is necessary to show measurements all taken at based temperature with an equilibrium temperature below T_c (error bar included) together with just above T_c (error bars included), so generally completing Fig.2 by reaching an equilibrium temperature above T_c (for which the antistoke SW should be gone).

We have clearly shown that the sample is in its superconducting state at our highest pump fluence (see above). In the normal state we can pump the system even harder and no new contribution to the signal appears. The temperatures we are giving are completely consistent with the picture that the Mexican-Hat potential has relaxed and Cooper Pairs are condensing. The temperatures as such are not constant even though the system can be regarded as thermalized. We argue that our system is thermalized since our fluence-dependent superconductivity-induced effects vanish at fluences of $139.3 \pm 28.5 \mu\text{J cm}^{-2}$ which is in line with our estimated heating by matching Stokes and anti-Stokes responses (The lower limit of the error bar is $111 \mu\text{J cm}^{-2}$. Our heating estimation results in a temperature around T_c at a fluence of $113 \mu\text{J cm}^{-2}$. The error bars of both independent methods overlap.).

- Also, the superconducting features extracted from the Stoke part at 113 and even at $75 \mu\text{J}\cdot\text{cm}^{-2}$ do not convince me that the compound is in the superconducting state. It is within the error bar of the intensity.

This interpretation does not accurately reflect the figures. The data points are clearly outside the error bars above zero, and the sample is clearly in its superconducting state at our highest pump fluence (see above). We validate this in our new Fig. S12 by analyzing the superconductivity-induced feature as a function of fluence compared to the normal state (100 K). The result shows, that the sample has a clear pair-breaking feature at our highest pump fluence, and based on a linear regression of the integrated data, a fluence of $139.3 \pm 28.5 \mu\text{J cm}^{-2}$ is necessary to reach T_c .

Some general comments:

The figures must include the equilibrium temperature.

Please explain in more detail the process used to obtain Fig S8, c) and say at what base temperature it was done.

Fig. S8 c) was done at a base temperature of 300 K. We repeated exemplary measurements at 100 K (see Fig. 1) and found the same heating rates at 100 K. This is shown in Fig. 1, where we can mirror the 100 K data between Stokes and anti-Stokes with the same thermal factors. Furthermore, we validate the derived temperatures (300 K) in the superconducting state (8 K) by analyzing the superconductivity-induced feature as a function of fluence compared to the normal state (100 K). This is shown in our new Fig. S12.

Figure S12: What is the meaning of the black line?

The black line in Fig. S12 represented the parameterization of the data according to all figures in the main text. We thank the reviewer for pointing out that this was missing in the caption

of Fig. S12. However, in our revised version, we removed Fig. S12 because it did not contain any new information and replaced it by a new figure analyzing the pair-breaking feature as a function of pump fluence.

Some references to the figures in the supplement need to be checked.

We have carefully reviewed the references to the figures in the SI and found no mistakes.

Reviewer #1 (Remarks to the Author):

In this manuscript, the authors introduced a new technique, Non-Equilibrium Anti-Stokes Raman Scattering (NEARS), to probe the non-equilibrium state of the high temperature superconductor Bi-2212 after pumping it with a 800nm laser pulse. They found extra anti-stokes signals around 25 meV below twice the quasi-particle gap. It cannot be explained by mapping the stokes signals with a Bose factor and a well-defined temperature. The authors attribute this signal to the Higgs mode excited by the pump, and claim to have directly observed the Higgs mode. The experiment is an interesting one that brings new blood to the ultrafast technique. The theoretical modeling, by itself, makes better sense than most analysis in previous experiments.

We thank the reviewer for this positive introductory assessment.

Nevertheless, the claim that this experiment has directly observed the Higgs mode appears to be an over-claim to me, especially considering that the peak is so broad (one should look at red area in Fig.2ab instead of Fig.2cd where the fitting curves are a little misleading). The other possibilities are not convincingly ruled out. The extra anti-stokes signal (Fig. 2cd) could be any nonthermal effect that the thermal Bose factor does not account for. For example, the other possibilities are:

We agree that the assignment has to be considered with care. However, a crucial aspect of our effect is that it emerges only below T_c in the superconducting state, which is essential for the Higgs mode. In contrast, other effects unrelated to superconductivity should also manifest above T_c . Furthermore, the broad nature of the Higgs mode is precisely what one would expect in a d-wave superconductor. [Varma, C.M. Higgs Boson in Superconductors. Journal of Low Temperature Physics 126, 901–909 (2002). <https://doi.org/10.1023/A:1013890507658>]

We revised the corresponding sentence in our introduction:

*„Experiments on s-wave superconductors **confirm** that Higgs modes are stable excitations,[20] whereas in d-wave superconductors **they** are metastable due to interactions with remanent nodal quasiparticles,[19, 22, 23] **resulting in a spectral broadening of the Higgs mode.**”*

We would expect to see sharper Higgs modes in s-wave superconductors. However, d-wave high- T_c superconductors are particularly well-suited for our measurements due to their large gap values. Moving forward, we plan to enhance our setup and investigate s-wave superconductors with lower gap values.

(A) It could be from the intra-band transition of excited quasi-particles which have a nonthermal distribution.

Intra-band transitions are low energy excitations within one band. The excited quasiparticles are exactly the quasiparticle continuum that we measure and from which we show that it is thermalized (see also Perfetti et al., Phys. Rev. Lett. 99, 197001 (2007)). The non-thermalized contribution occurs only below T_c as stated above.

(B) It could be the Bardasis-Schrieffer mode that is excited nonlinearly by the pump, similar to the Higgs mode. The authors made some arguments in SI (S7) against the Bardasis-Schrieffer mode which are not convincing: “However, the Bardasis-Schrieffer mode would leave a fingerprint on the Stokes side,

as observed in pnictides. We see no indication of this mode". The same argument would apply equally well to the story of Higgs mode.

The Bardasis-Schrieffer mode describes an excitation of a subdominant pairing channel. It is challenging to detect experimentally and has been discussed in pnictides [Kretzschmar et al., Phys. Rev. Lett. 110, 187002 (2013); Böhm et al., npj Quant Mater 3, 48 (2018)]. There is no evidence for the existence of a Bardasis-Schrieffer mode in cuprates. This mode strongly depends on the relative coupling of the pairing channels. In a transient spectroscopic experiment, it is possible to excite the Bardasis-Schrieffer mode via the pump; however, a strong fluence dependence of the Mexican-Hat potential is expected (with the subdominant pairing channel in Bi-2212 assumed to be of s-wave character) [Sun et al., Phys. Rev. Research 2, 023413 (2020)].

Furthermore, this mode would need to be metastable over several picoseconds to be overpopulated on the anti-Stokes side in NEARS. Contributions on both the Stokes and anti-Stokes sides would be accounted for in the NEARS analysis. In summary, the Bardasis-Schrieffer mode is highly unlikely to appear as a dominant feature in NEARS in cuprates. The observation of an almost constant energy as a function of fluence contradicts the expectations for the Bardasis-Schrieffer mode but aligns with those for a Higgs mode in d-wave superconductors.

We have changed the discussion in the SI accordingly and thank the reviewer for his/her comment:

*"Bardasis-Schrieffer mode: This mode represents a subdominant pairing channel that could be activated by pumping the SC state. Its excitation energy would be below the binding energy 2Δ of the dominant pairing channel and this mode would also not be subject to screening. The Bardasis-Schrieffer mode would leave a **distinct signature** on the Stokes side, as observed in pnictides.[25, 26, 27] **However, our Stokes data shows no evidence of this mode. Moreover, our analysis, which relies on comparison of Stokes and anti-Stokes intensity, inherently accounts for its potential presence. Furthermore, to the best of our knowledge, there is no evidence of Bardasis-Schrieffer modes in cuprates. In a pump-probe experiment, a pump-activated Bardasis-Schrieffer mode would typically shift to lower energies with increasing pump fluence.[Sun et al., Phys. Rev. Research 2, 023413 (2020)] However, we observe that the energy of the NEARS feature remains nearly constant across different fluences. In addition, NEARS is driven by population inversion, which amplifies the anti-Stokes signal relative to the Stokes signal (see Fig. S9). For this to occur, the observed mode must be metastable over several picoseconds. This serves as a strong argument for the Higgs mode's relevance in explaining our observations, as its metastable nature facilitates population inversion. Additionally, all our data and in particular of the SC gap feature confirm that we remain in the superconducting state. This leaves the Higgs mode as the best explanation for the NEARS feature."***

Furthermore, given that the width of the mode is about 20 meV, it should decay within 0.2 ps into incoherent degrees of freedom, mostly quasi-particles. One should not expect to see it at a delay of 3ps. More discussion on the dependence of pump-probe delay would help identify the origin of the extra anti-stokes signal.

In a non-equilibrium, transient system, it is challenging to directly correlate width with lifetime. Two key factors must be considered:

First, even in the steady state, the mode width and energy are influenced by different contributions. There is lifetime and energy, but there are also vertex corrections. In our case, vertex corrections are very important. As we vary the parameter r in our calculations, both the peak-width and energy shift accordingly, as illustrated in Fig. R1. This behavior is analogous to a phonon with vertex corrections,

leading to the Fano anti-resonance effect, where the phonon's position and width do not directly correspond to its true energy or lifetime, too.

Second, the temporal width of the probe and pump lasers (both 1.2 ps) imply that we are likely sampling different Mexican-Hat potentials throughout the relaxation process. The Mexican-Hat potential is not static but exists in a transient state, resulting in varying eigenfrequencies of the amplitude mode at different moments in time. Thus, we measure a signal broadened by the time-average. Additionally, the pair-breaking peak has not yet fully recovered (see Fig. 2 and Fig. S12), further supporting this interpretation. Consequently, experimental limitations would contribute to broadening the measured Higgs mode.

Fig. R1: Higgs response calculated as explained in detail in SI S.2 and presented in Fig. 4 of the main text. Here: direct comparison between $r = 0$ and $r = 0.9$ to demonstrate the influence of the vertex correction of energy and lifetime of the observed mode.

Without the strong claim of the Higgs mode, this work could be a solid one that serves as a benchmark for the technique and for further measurements. Therefore, I would suggest being conservative and claiming that this experiment has only seen signatures of the Higgs mode. The other possibilities should be briefly discussed too.

We adjusted the wording in the title and other sections accordingly and considered additional possibilities in the main text, in addition to section S.7 in the SI.

Most relevant changes made in our revised manuscript (see also manuscript with marked changes):

Title: ***“Non-Equilibrium Anti-Stokes Raman Spectroscopy for Investigating Higgs Modes in Superconductors”***

Abstract: ***“We report on an innovative spectroscopic technique to study symmetries and energies of the Higgs modes in the high-temperature superconductor Bi-2212 after a “soft quench” of the Mexican-Hat potential. Population inversion induced by an initial laser pulse, leads to an additional anti-Stokes Raman-scattering signal, which is consistent with polarization-dependent Higgs modes.”***

Here we introduce a new spectroscopic technique: Non-Equilibrium Anti-Stokes Raman Scattering (NEARS). NEARS utilizes a so-called soft quench of the Mexican-Hat potential, as we will detail below, with the goal of populating Higgs modes of different symmetries, which are then probed by anti-Stokes Raman scattering.

“Conventional Raman scattering excites quasiparticles leading to energy-loss spectroscopic features on the Stokes side (see Fig. 1a-d). In superconductors, this technique is sensitive to low-energy excitations, such as PB excitations[33, 34], see Fig. 1c and d, density-correlation functions of Josephson plasmons[31], Leggett modes[35], and Bardasis-Schrieffer modes[36, 37]. Our aim is to measure the relaxation of a superconductor and the concomitant population of Higgs modes in the quasi-static limit and in nearly thermal equilibrium. As a result, excitations that are not the lowest-energy metastable

states will decay more rapidly than the Higgs mode and thus remain undetectable in NEARS. This strongly supports the Higgs mode's relevance in interpreting any additional features observed in NEARS measurements."

Some other comments on the writing:

(A) "Higgs particle" is an unnecessary terminology. No quantum properties of the Higgs mode is needed to explain the data. Coherent oscillation of the mode or its thermal fluctuation could totally give the signal that the experiment observes. Therefore, I would suggest a more appropriate name, e.g., Higgs mode.

We switched the labeling from "particle" to "mode".

(B) I don't understand why the Higgs mode is "metastable".

The Higgs mode represents the lowest-energy excitation in the two-particle channel. In an s-wave superconductor, it is a sharp, well-defined excitation that remains stable since it cannot decay into other modes. However, in a d-wave superconductor, the nodes of the superconducting order parameter introduces momentum-dependent decay channels, rendering the Higgs mode metastable and leading to the increased linewidth compared to the s-wave case, as the reviewer mentioned himself.

(C) In the end, it says "Meissner effect requires the presence of the Higgs field" which I cannot agree with. For example, a Bose-Einstein condensation type of superconductor does not have a Higgs mode.

A Bose-Einstein Condensate itself carries no charge. It is the presence of charge that provides coupling to the gauge field necessary for the Anderson-Higgs mechanism. While relativistic Bose-Einstein Condensates can exhibit amplitude modes, they do not support Higgs modes. As discussed by Anderson [P. Anderson, Phys. Rev. 110, 985 (1958), and Phys. Rev. 110, 827–835 (1958)], the symmetry breaking of a charged field leads to two simultaneous effects: first, the emergence of the Meissner effect, and second, the phase mode being pushed up to the plasma frequency, leaving the Higgs mode as the lowest excitation in the two-particle channel. This establishes a direct correspondence between the presence of a Higgs mode and the Meissner effect. Consequently, while Bose-Einstein Condensates may exhibit amplitude modes, they do not host Higgs modes which are the amplitude modes of a charged field that gives mass to gauge bosons associated with that charge i.e. a field giving rise to the Higgs mechanism. Although the referee may find our language somewhat picky, we consider it crucial, as the analogy between the Higgs mode in superconductors and High-Energy Physics is only valid in the presence of a gauge field.

Reviewer #2 (Remarks to the Author):

The authors present a non-equilibrium Raman scattering investigation of Bi-2212 superconductors, identifying an additional spectral feature in the anti-Stokes region of the Raman response as a Higgs mode signature following a soft quench of the superconducting condensate. The detection of the Higgs mode holds fundamental significance in condensed matter physics. Prior studies have largely focused on probing this mode via THz-field-driven third-order susceptibility. This work introduces a new

perspective by employing non-equilibrium anti-Stokes Raman scattering, offering a distinct experimental approach to Higgs mode detection.

Thank you very much.

The experimental data is technically sound, analyzed properly, and presented in sufficient detail. I agree with previous referees. To further solidify the interpretation linking the 25 meV signal to the Higgs mode, extending measurements to higher pump fluences beyond $113 \mu\text{J}/\text{cm}^2$ —thereby demonstrating signal suppression upon transition to the normal state at the base temperature (8 K)—would strengthen the argument. This might not be easy as the authors noted in the reply that their current setup limits the highest pump fluence to $113 \mu\text{J}/\text{cm}^2$. However, such investigations could be pursued in future studies.

Thank you. We will certainly explore it further in future studies.

Overall, I think it is an excellent paper. The experimental progress is remarkable and the experimental results are interesting, and the content of this paper is suitable for the broad audience of Nature Communications. The findings presented in this paper motivate the future non-equilibrium Raman scattering investigation of Higgs mode in more unconventional superconductors such as iron-based superconductors and other multiband superconductors.

We sincerely appreciate the reviewer's valuable feedback. We look forward to exploring different types of superconductivity further in future studies.

Reviewer #3 (Remarks to the Author):

I have read with great interest the paper by Glier et al. on the direct observation of a Higgs mode via non-equilibrium Raman scattering. While there have been several reports of Higgs modes in high- T_c cuprates—some more direct or convincing than others—I tend to approach these claims with skepticism, as I rarely find the experimental data truly compelling. More often than not, interpretations seem biased toward the narrative the authors wish to support. Reading this work in detail, I began with mixed feelings that gradually developed into serious concerns, ultimately preventing me from recommending its publication in this or any other journal unless a major issue is addressed.

We conceptually understand the concerns and the bias of the referee and to some extent, these are also shared by us.

In short, while I find certain aspects of the Higgs mode discussion convincing (which I will return to), I have significant doubts regarding key elements of the equilibrium data—doubts with serious implications. Specifically, I am surprised by the apparent lack of selection rules for the Raman-active phonons. The A_{1g} and B_{1g} spectra appear suspiciously similar, and the presence of an intense mode at 60 meV in the B_{1g} spectrum immediately caught my attention. I do not recall seeing such a feature in the extensive Raman literature on Bi2212 over the past two decades (e.g., Sugai et al., PRB 2003; Loret, PRB 2017).

In our manuscript, we cite the work of Budelmann et al. (2005) [Budelmann, D. et al., Phys. Rev. Lett. 95, 057003 (2005)]. It is important to recognize that the Raman matrix element depends on the incident photon energy used in the experiment, leading to an amplification of the 2Δ excitation observed in the Raman data. Most Raman setups operate only in the visible to near-visible range, whereas our instrument is unique - it is fully achromatic, utilizing a fully reflective entrance objective [Schulz, B. et al., Rev. Sci. Instrum. 76, 073107 (2005)], and functions across a spectral range from 1000 nm to 200 nm. Thus, there is no discrepancy with existing data. Rather, the limitations of conventional Raman instruments, which typically operate only in the visible range, explain the scarcity of data beyond this range.

The UV spectral range allows for the detection of stronger signals from pair-breaking excitations and phonons while ensuring that the probe volume is fully pumped, as the 800 nm pump beam is larger and penetrates deeper into the sample than the 400 nm probe beam. To prevent jitter between the pump and probe in time-resolved experiments, we use ω and 2ω . Reversing the pump and probe wavelengths would introduce several issues: the higher probe spectral resolution at 800 nm would extend the measurement time by a factor of four, and using a higher-energy pump than the probe would induce additional Raman scattering on its Stokes side due to excitons [Salamon, D. et al., Phys. Rev. B 53, 886 (1996)]. The Raman matrix element for pair-breaking excitation depends on the incident photon energy, effectively mapping different regions in k-space as the photon energy changes. Therefore, the observed effects in pair-breaking excitations are expected and well accounted for by theory (see section S.2).

Bi2212 is structurally complex, leading to a rich phonon spectrum that is not yet fully understood. However, in most previous studies, selection rules hold to a reasonable extent, with many A_{1g} features suppressed in crossed polarization channels probing the B_{1g} or B_{2g} responses. This does not seem to be the case here, which is troubling, as it undermines the symmetry-based interpretation of the Higgs mode.

Budelmann et al. conducted a detailed study of the resonance Raman spectra of Bi-2212 in 2005, published in Physical Review Letters, demonstrating that the spectra exhibit a strong dependence on excitation energy, particularly when tuning into the UV spectral range (as seen in the 3.41 eV dataset in the figure below). In their findings, disorder-activated phonon modes appear in the B_{1g} symmetry. However, the majority of Raman studies [e.g., Sugai et al., PRB 2003; Loret et al., PRB 2017] have been performed in the visible spectral range (1.8–2.5 eV).

FIG. 2 (color). Experimental data (wiggled lines) and the model (solid line) of the electron and phonon responses as a function of the incident photon energy between 1.97 and 3.53 eV. In column (a) the bold solid line represents a fit to the experimental data. Also shown as thin solid and dashed lines are the electronic contribution due to the gap features at 320, 440, and 580 cm^{-1} as well as the marginal Fermi liquid, respectively. (b) Changes of the shape in the electronic response as a function of incident photon energy. Responses have been scaled to match.

[Budelmann, D. et al., *Phys. Rev. Lett.* 95, 057003 (2005)]

We have added an additional chapter (S.8) and figure to the Supplementary Information (SI). Fig. S10 b) (see below and revised version of SI) shows Stokes intensities (probe only) at 10 K for all three measured symmetry configurations. A clear suppression of several phonon modes is observed in predominantly B_{1g} (green) and B_{2g} (red) symmetries compared to predominantly A_{1g} symmetry (blue). As demonstrated by Budelmann et al., disorder-induced phonon modes emerge as a function of incident photon energy when probing charge-transfer excitation in Bi-2212.

Fig. S10: a) Raman susceptibilities of the pair-breaking feature in the experimental scattering configurations $A_{1g}+B_{2g}$ (blue), $A_{2g}+B_{1g}$ (green), and $A_{2g}+B_{2g}$ (gray). To extract these curves from our Raman intensities, we corrected for the Bose function and subtracted all phonons as determined by our parameterization approach (see eq. S30). The inset shows the comparison between the electronic B_{1g} feature (green, $A_{2g}+B_{1g}$) together with the subtracted feature $A_{1g}+B_{2g} - A_{2g}+B_{2g}$ (dark blue). b) Stokes intensities (probe only) at 10 K for all three measured symmetry configurations. A clear suppression of phonon modes is observed in predominantly B_{1g} (green) and B_{2g} (red) symmetries compared to predominantly A_{1g} symmetry (blue). As demonstrated by Budelmann et al.[11], disorder-induced phonon modes emerge as a function of incident photon energy when probing charge-transfer excitation in Bi-2212.

The idea that the extra intensity in the pumped anti-Stokes response may be linked to the Higgs mode of the superconducting condensate is plausible. I share some of the other reviewers' concerns regarding potential pump-laser heating, but I found this issue to be carefully discussed in the manuscript, leading to a relatively convincing case—certainly more so than many previous studies published in prestigious journals, which were far less rigorous.

Thank you for pointing this out and agreeing with our analysis.

However, if the selection rules for phonons cannot be convincingly demonstrated, and if modes expected to be inactive in the B_{1g} channel appear prominently, how can one confidently attribute the anti-Stokes intensity in this channel to a true B_{1g} symmetry response? It seems evident that the B_{1g} spectrum suffers from significant polarization leakage from the A_{1g} channel. This issue is not limited to the 60 meV mode but affects all A_{1g} features, which appear with substantial intensity in the B_{1g} spectrum. If such leakage occurs at the phonon level, it is highly likely that any other A_{1g} feature—including the suspected Higgs mode and the pair-breaking response—also contaminates the B_{1g} spectrum.

Firstly, we have pointed out how disorder-induced phonon modes emerge as a function of incident photon energy when addressing charge-transfer excitation in Bi-2212, a phenomenon demonstrated two decades ago [Budelmann, D. et al., Phys. Rev. Lett. 95, 057003 (2005)]. The superconductivity-induced features originate from the copper-oxygen planes and the orthorhombic distortion, which mixes B_{1g} and A_{1g} symmetries - an effect that has been well known for a long time. Our pump geometry excites both A_{1g} and B_{1g} Higgs modes, and this is precisely what we observe. Notably, we do not detect a B_{2g} Higgs mode (see Fig. S11). Therefore, the question of whether there is an A_{1g} contribution in the B_{1g} Higgs mode, and vice versa, is not critical to our claim.

For the symmetry selection rules of the phonons, we refer to our previous response and added chapter S.8 to our SI.

This is the central issue: polarization selection rules are a fundamental strength of Raman scattering, and phonons provide an excellent benchmark for their validity. If the phonons themselves do not adhere to these rules, it is not possible to use symmetry arguments for the electronic excitations.

We have addressed these concerns. Unfortunately, phonons do not serve as a precise benchmark for symmetry selection rules in Bi-2212. However, we have demonstrated that the symmetry selection rules hold approximately, and, most importantly, that the relevant rule governing the presence of the Higgs mode in A_{1g} and B_{1g}, but not in B_{2g}, remains valid.

Given the strong claims made regarding the symmetry of the Higgs mode, I see this as a critical weakness of the study. Unless this issue is properly addressed, I cannot recommend the manuscript for publication.

We have addressed the referee's concerns, which can be attributed to the intrinsic orthorhombic distortion of the material and disorder-induced phonon modes in the buffer layers of Bi-2212. These effects have been extensively studied in the literature [Munnikes, N. et al., Phys. Rev. B 84, 144523 (2011); Hackl, R. et al., Journal of Physics and Chemistry of Solids 67, 289–293 (2006); Boulesteix, C., et al., Journal of Physics: Condensed Matter 12, 9637 (2000)]. The key symmetry selection rule remains that Higgs modes are activated in A_{1g} and B_{1g} symmetries but not in B_{2g} (see Fig S11).

Here are a few additional comments:

- The introduction is tedious to read. It contains all relevant information but this succession of short and often disconnected sentences can be hard to follow.

Thank you for the feedback. We finetuned the reading flow in our introduction and refer to the revised version of our manuscript with marked changes.

- The sketches in Fig. 1a are misleading. The blue arrows should have the same length as in the equilibrium spectra the energy of the incident photon is presumably fixed (if not then resonance effects should be considered)

We appreciate the reviewer's comment. However, the schematic in Fig. 1a is not drawn to reflect the actual energy scales. The incident laser energy is in the eV range, while the Raman shift energy range is in meV. A true-to-scale representation would not be practical for this figure. To prevent any confusion, we have revised the figure caption accordingly:

*“(a) Energy diagrams of Stokes and anti-Stokes Raman scattering. The scattering cross-sections of equilibrium Stokes (energy-loss) and anti-Stokes (energy-gain) scattering are linked by the Bose function. The Raman probe excitation is depicted in blue, the scattered light is shown in black. **Please note that this schematic is not drawn to scale. In all our measurements (Stokes and anti-Stokes), the Raman probe excitation (blue) is 3 eV, while the Raman shift falls within the meV range.**”*

- The anti-Stokes function at 8K is not 'effectively zero' but I agree it is weak (vanishingly small would be more appropriate)

Thank you, we changed the sentence in the figure caption:

*“The anti-Stokes Raman spectrum at 8 K is vanishing because the anti-Stokes Bose-function at 8 K is **infinitesimally small.**”*

- How can the electronic background be 'constant' at 100K, shouldn't it go to 0 at 0 energy as any response function???

It indeed approaches zero at zero energy, which is why the background is fitted with a tanh function. Our results show that for $T > T_c$, the critical frequency ω_c (which accounts for the gap opening in the superconducting state) tends to zero, resulting in an effectively constant background at finite energies (> 0). We appreciate you bringing this to our attention and will revise the wording in the figure caption accordingly. However, we have always provided a detailed explanation of the procedure for fitting the electronic background (see section S.6 of initial submission).

Changes made in caption of Fig. 1:

*“At 100 K the electronic background is constant (black solid line) **within the presented energy range. The applied tanh function with a small ω_c approaches zero at Raman shifts below 10 meV**”*

- Given the complexity of the spectra, the fitting details should be presented, not just sketches.

We are unsure which sketches are being referred to. However, we agree with the reviewer that the fitting procedure is important, and we reference the relevant detailed chapter in the supporting information (see section S.6 of initial submission).

- A proper discussion on selection rules of phonons is missing

We added chapter S.8 and Fig. S10 (b) to our SI:

S.8 Raman Selection Rules

Disorder and orthorhombic distortions play a significant role in shaping the Raman spectra and selection rules of Bi-2212.[28, 29, 30] Disorder, particularly in the buffer layers, can induce additional Raman-active phonon modes, modifying spectral features and enhancing anisotropic charge fluctuations. This disorder-induced phonon behavior is further influenced by doping, as phonon modes in Bi-2212 shift due to variations in carrier concentration, leading to spectral weight enhancement in specific symmetries.[28] Orthorhombic distortions break the ideal tetragonal symmetry and modify Raman selection rules by lifting degeneracies and enabling new phonon activations as also seen by resonance Raman studies.[11] Most importantly, resonance Raman studies demonstrate the appearance of disorder-induced phonon modes in the UV spectral range close to the charge-transfer transition and in general the strong dependence of the Raman matrix element on incident photon energy, leading to an amplification of the 2Δ excitation.[11] Fig. S10 b) shows the Raman spectra for all three measured symmetry configurations. We observe a clear phonon suppression when comparing $A_{2g}+B_{1g}$ and $A_{2g}+B_{2g}$ with $A_{1g}+B_{2g}$ data demonstrating that several phonons have A_{1g} character. This symmetry-dependent behavior of phonons in Bi-2212 in the UV spectral range is in line with previous observations.[11] Most importantly, our measurements confirm the expected selection rules for the Higgs modes. In this experimental configuration, the Higgs mode is predicted to be excited in the A_{1g} and B_{1g} symmetries but not in the $A_{2g} + B_{2g}$ configuration [31], as demonstrated in Fig. S11.

Fig. S10: a) Raman susceptibilities of the pair-breaking feature in the experimental scattering configurations $A_{1g}+B_{2g}$ (blue), $A_{2g}+B_{1g}$ (green), and $A_{2g}+B_{2g}$ (gray). To extract these curves from our Raman intensities, we corrected for the Bose function and subtracted all phonons as determined by our parameterization approach (see eq. S30). The inset shows the comparison between the electronic B_{1g} feature (green, $A_{2g}+B_{1g}$) together with the subtracted feature $A_{1g}+B_{2g} - A_{2g}+B_{2g}$ (dark blue). b) Stokes intensities (probe only) at 10 K for all three measured symmetry configurations. A clear suppression of phonon modes is observed in predominantly B_{1g} (green) and B_{2g} (red) symmetries compared to predominantly A_{1g} symmetry (blue). As demonstrated by Budelmann et al.[11], disorder-induced phonon modes emerge as a function of incident photon energy when probing charge-transfer excitation in Bi-2212.

Point-by-Point Reply to the Reviewer's Comments:

Reviewer #1 (Remarks to the Author):

The authors made an effort to address my concerns. Most of them make sense except the following two:

A: Regarding the Bardasis-Schrieffer mode, the authors wrote "In a pump-probe experiment, a pump-activated Bardasis-Schrieffer mode would typically shift to lower energies with increasing pump fluence.[Sun et al., Phys. Rev. Research 2, 023413 (2020)] However, we observe that the energy of the NEARS feature remains nearly constant across different fluences. In addition, NEARS is driven by population inversion, which amplifies the anti-Stokes signal relative to the Stokes signal (see Fig. S9). For this to occur, the observed mode must be metastable over several picoseconds. This serves as a strong argument for the Higgs mode's relevance in explaining our observations, as its metastable nature facilitates population inversion. Additionally, all our data and in particular of the SC gap feature confirm that we remain in the superconducting state. This leaves the Higgs mode as the best explanation for the NEARS feature."

I don't see why "a pump-activated Bardasis-Schrieffer mode would typically shift to lower energies with increasing pump fluence" while the same thing does not happen to the Higgs mode. Increasing pump fluence heats up the sample and reduces the gap, which could totally shift all the modes. Furthermore, I don't see a good reason that Higgs mode could be metastable while the Bardasis-Schrieffer mode cannot be. From my understanding, none of the forementioned features is a property of the Higgs mode not shared by the Bardasis-Schrieffer mode.

We thank the reviewer for pointing this out. It is important to emphasize that the activation of the Higgs mode involves not just heating, but an actual change in (over)population. Heating effects have been explicitly addressed in our manuscript. In contrast to heat (i.e. effective temperature), population inversion is a non-equilibrium concept that is rooted in quantum physics governed by lifetimes of the involved states. In order to occur, a meta-stable mode needs to be filled up by optical activated fast lived states. Assuming an s-wave gap for simplicity, the Higgs mode does not decay, while the Bardasis-Schrieffer mode does. This conceptual difference will be maintained also for different symmetries of the order parameter. The energy of the Higgs mode is determined by the curvature of the Mexican Hat potential near its minimum. Therefore, after relaxation of the free-energy landscape (expected to occur within less than 1 ps) the mode energy remains constant, and only its population can be modified.

In contrast, a Bardasis-Schrieffer mode is associated with a different Mexican Hat potential, which does not correspond to the free-energy minimum but rather lies at a higher energy. Even if such a mode is transiently excited, it is expected to relax toward the true ground state at a rate proportional to its energy offset and, therefore, does not constitute a long-lived metastable excitation. Only in the limiting case where both potentials are nearly degenerate, the Bardasis-Schrieffer mode might exhibit a behavior resembling that of the Higgs mode. However, after more than 30 years of research on high-temperature superconductors, there is no compelling evidence supporting the presence of Bardasis-Schrieffer modes in cuprates contrary to the pnictides.

We have added a note in the SI to acknowledge this special case:

In the presence of nearly degenerate pairing channels, a metastable Bardasis-Schrieffer mode may, in principle, arise [Sun et al., Phys. Rev. Research 2, 023413 (2020)]. Nonetheless, to the best of our knowledge, and despite more than three decades of research on high- T_c superconductors, there is still no convincing evidence supporting such a scenario in cuprates.

B: Regarding whether “Meissner effect requires the presence of the Higgs field”, what I meant is a Bose-Einstein condensation (BEC) type of superconductor made of charged pairs of electrons. This has been suggested by some researchers to be the case of High T_c superconductors. It of course couples to the electromagnetic field and its phase mode is shifted to be the plasmon. It also exhibits the Meissner effect. However, it does not have an amplitude mode (Higgs mode) because the amplitude and phase fluctuations are a conjugate pair of the same mode, i.e., the phase mode. In this case, I don’t think it is appropriate to call the order parameter of a BEC type of superconductor a “Higgs field”, as opposed to the BCS case whose analogy to the relativistic Higgs field is more natural. Therefore, I strongly disagree with the statement that “Meissner effect requires the presence of the Higgs field”.

We thank the reviewer for the clarification. The referee might refer to the literature in the 50’ties where the question how and if a charged BEC can exhibit a Meissner effect was hotly debated (e.g. M.R. Schaffroth Phys. Rev. 100, 463 (1955); J. Bardeen, L.N. Cooper, J.R. Schrieffer Phys. Rev. 108, 1175 (1957); P.W. Anderson Phys. Rev. 110, 827 (1958)) in particular with respect to gauge invariance, transversal and longitudinal excitations in superconductors as well as back-flow corrections to the vertex. Some authors have introduced this concept more recently (Koh PRB 68, 144 502 (2003)).

We all agree that the BCS Hamiltonian can be rewritten in quasi-relativistic form as outlined in our manuscript. The non-relativistic Hamiltonian would be a Gross-Pitaevskii Hamiltonian. The latter does not allow to gauge away the phase mode after introducing the electromagnetic field tensor and the Anderson-Higgs mechanism is not possible. The equations of motion for the electromagnetic field do not yield an obvious Meissner effect, which does not mean that effects similar to the Meissner effect are completely impossible. We refer to the appendix R1 to our response appended below with respect to the bosonic two-particle channel.

In short, in a charged BEC an effect similar to the Meissner Effect is only possible if one introduces additional residual interactions in the BEC Hamiltonian. Please see S. Koh, Phys. Rev B 68 144502 (2003) – in particular see page 7 of the article discussion point (ii) where Koh states the difference between the two different “Meissner” effects and the relevance of vertex corrections to the current-current response tensor in an BEC that are not necessary in a BCS Hamiltonian. This makes this type of “Meissner Effect” vulnerable to further interactions that will most likely remove the singularity in the current-current response tensor quenching the Meissner effect in a charged BEC.

Given the complexity of this topic we have changed the corresponding sentences in the main text for clarification:

“Consequently, the Meissner effect signifies a macroscopic quantum condensate in which a photon acquires mass, representing a one-to-one analogy to high-energy physics.”

“Since the Meissner effect requires the presence of the Higgs field, the observation of a Higgs mode can serve as a novel criterion for superconductivity.”

to

*“Consequently, the **BCS-like** Meissner effect signifies a macroscopic quantum condensate in which a photon acquires mass, representing a one-to-one analogy to high-energy physics.”*

*“Since the **BCS-like** Meissner effect requires the presence of the Higgs field, the observation of a Higgs mode can serve as a novel criterion for superconductivity.”*

Reviewer #2 (Remarks to the Author):

I recommend that the manuscript be published in Nature Communications.

We appreciate the reviewer's positive feedback.

Reviewer #3 (Remarks to the Author):

I thank the authors for their thorough and thoughtful response to my concerns, particularly with regard to the selection rules. Bi2212 is indeed a structurally complex material, and the influence of resonance effects—especially on phonons and gap excitations—can be quite pronounced. I must acknowledge that I had previously assumed a more canonical behavior in this regard, and I appreciate the authors for drawing my attention to the relevant literature.

That said, I cannot entirely overlook the fact that resonance-induced activation of defect modes and significant alterations in the superconducting electronic response raise important questions. Specifically, such effects appear to blur the applicability of selection rules, which complicates the interpretation. While I recognize the value and quality of the experimental data presented, I remain skeptical of the proposed interpretation. It would perhaps be too demanding to request a full resonant study of the presumed Higgs modes for this otherwise excellent experimental work. Nonetheless, it seems reasonable to ask to what extent the current interpretation depends on these resonance effects.

With this caveat in mind, I believe the manuscript presents novel and intriguing data that merit publication. The results are likely to stimulate valuable discussion within the community, and Nature Communications appears to be an appropriate venue for such work.

We thank the reviewer for the insightful comments. We agree that a resonance study would be very valuable. It is one important experimental pathway that we will certainly follow, but beyond the scope of the current manuscript as the reviewer also agrees.

Anderson-Higgs in Superconductors:

PHASE Transition & SYMMETRY Breaking

Spontaneous Symmetry Breaking (SSB)

- In the normal state: Ground-state has the same U(1) symmetry than the potential
- In the SC state: New ground-state has lower symmetry → spontaneous symmetry breaking

Let us consider an approach with **Amplitude and Phase fluctuations:**

Lagrangians of the condensate:

$$\mathcal{L}^{KG} = (\partial_\mu \psi)^* (\partial^\mu \psi) - V(\psi)$$

$$= (\partial_t \psi)^* (\partial_t \psi) - (\nabla \psi)^* (\nabla \psi) - V(\psi)$$

→ Klein Gordon (Lorentz-invariant)

$$\mathcal{L}^{GP} = \psi^* (\partial_t \psi) - (\nabla \psi^*) (\nabla \psi) - V(\psi)$$

→ (non-Lorentz invariant) Gross-Pitaevskii (BEC)

$$\mathcal{L}^{KG} = (\partial_t H)^2 - (\nabla H)^2 + \psi_0^2 \left((\partial_t \theta)^2 - (\nabla \theta)^2 \right) + 2\alpha H^2$$

Euler-Lagrange

Equations of motions for amplitude and phase are decoupled and solved by plane waves.

$$(\partial_t^2 - \nabla^2)H = 2\alpha H$$

$$(\partial_t^2 - \nabla^2)\theta = 0$$

$$\omega_H^2 = q^2 - 2\alpha$$

$$\omega_\theta^2 = q^2$$

Amplitude (massive)

Phase (massless)

Anderson-Higgs in Superconductors:

PHASE Transition & SYMMETRY Breaking

Spontaneous Symmetry Breaking (SSB)

- In the normal state: Ground-state has the same U(1) symmetry than the potential
- In the SC state: New ground-state has lower symmetry → spontaneous symmetry breaking

Let us consider an approach with **Amplitude and Phase fluctuations:**

Lagrangians of the condensate:

$$\mathcal{L}^{KG} = (\partial_\mu \psi)^* (\partial^\mu \psi) - V(\psi)$$

$$= (\partial_t \psi)^* (\partial_t \psi) - (\nabla \psi)^* (\nabla \psi) - V(\psi)$$

→ Klein Gordon (Lorentz-invariant)

$$\mathcal{L}^{GP} = \psi^* (\partial_t \psi) - (\nabla \psi^*) (\nabla \psi) - V(\psi)$$

→ (non-Lorentz invariant) Gross-Pitaevskii (BEC)

$$-i\psi_0(\partial_t \theta) - (\nabla^2 H) = 2\alpha H$$

$$i(\partial_t H) - \psi_0(\nabla^2 \theta) = 0$$

Equations of motions for amplitude and phase are coupled and solved by plane waves.

$$H(\mathbf{r}, t) = \theta(\mathbf{r}, t) = e^{i(\omega t - \mathbf{q}\mathbf{r})}$$

Only one single mode without mass.

$\omega^2 = 2\alpha q^2 - q^4$

Amplitude (massive)

Phase (massless)

What if we charge the condensate ?

Anderson-Higgs in Superconductors:

PHASE Transition & SYMMETRY Breaking

Spontaneous Symmetry Breaking (SSB)

- In the normal state: Ground-state has the same U(1) symmetry than the potential
- In the SC state: New ground-state has lower symmetry → spontaneous symmetry breaking

Let us consider an approach with **Amplitude and Phase fluctuations:**

Lagrangians of the condensate with an electromagnetic field tensor:

$$\mathcal{L}^{KG} = (\partial_\mu \psi)^* (\partial^\mu \psi) - V(\psi)$$

$$= (\partial_t \psi)^* (\partial_t \psi) - (\nabla \psi)^* (\nabla \psi) - V(\psi)$$

→ Klein Gordon (Lorentz-invariant)

$$\mathcal{L}^{GP} = \psi^* (\partial_t \psi) - (\nabla \psi^*) (\nabla \psi) - V(\psi)$$

→ (non-Lorentz invariant) Gross-Pitaevskii (BEC)

Adding the photons !!

$$\mathcal{L}^{KG} = (D_\mu \psi)^* (D^\mu \psi) - V(\psi) - \frac{1}{4} F_{\mu\nu} F^{\mu\nu}$$

$$D_\mu = \partial_\mu + ieA_\mu$$

$$F_{\mu\nu} = \partial_\mu A_\nu - \partial_\nu A_\mu$$

$$\mathcal{L}^{KG} = \left(\partial_\mu H - ie \left(A_\mu + \frac{1}{e} \partial_\mu \theta \right) (\psi_0 + H) \right) \left(\partial^\mu H + ie \left(A^\mu + \frac{1}{e} \partial^\mu \theta \right) (\psi_0 + H) \right) + 2\alpha H^2 - \frac{1}{4} F_{\mu\nu} F^{\mu\nu}$$

Gauge transformation:

$$\psi' = \psi e^{-ie\theta}$$

$$A'_\mu = A_\mu + \frac{1}{e} \partial_\mu \theta$$

$$\mathcal{L}^{KG} = (\partial_\mu H) (\partial^\mu H) + 2\alpha H^2 - \frac{1}{4} F_{\mu\nu} F^{\mu\nu} + e^2 \psi_0^2 A_\mu A^\mu + 2e^2 \psi_0 A_\mu A^\mu H$$

Anderson-Higgs in Superconductors:

PHASE Transition & SYMMETRY Breaking

Spontaneous Symmetry Breaking (SSB)

- > In the **normal state**: Ground-state has the same U(1) symmetry than the potential
- > In the **SC state**: New ground-state has lower symmetry → spontaneous symmetry breaking

Let us consider an approach with **Amplitude and Phase fluctuations**:

Lagrangians of the condensate with an electromagnetic field tensor:

$$\begin{aligned} \mathcal{L}^{KG} &= (\partial_\mu \psi)^* (\partial^\mu \psi) - V(\psi) \\ &= (\partial_t \psi)^* (\partial_t \psi) - (\nabla \psi)^* (\nabla \psi) - V(\psi) \\ \mathcal{L}^{GP} &= \psi^* (\partial_t \psi) - (\nabla \psi^*) (\nabla \psi) - V(\psi) \end{aligned}$$

- Klein Gordon (Lorentz-invariant)
- (non-Lorentz invariant)
- Gross-Pitaevskii (BEC)

Adding the photons !!

$$\mathcal{L}^{KG} = (\partial_\mu H)(\partial^\mu H) + 2\alpha H^2 - \frac{1}{4} F_{\mu\nu} F^{\mu\nu} + e^2 \psi_0^2 A_\mu A^\mu + 2e^2 \psi_0 A_\mu A^\mu H$$

Mass term for the photons. If there is no charge it is zero even though the orderparameter can be finite!

Note the quadratic coupling to the Higgs → Raman scattering! No linear coupling

$$(\partial_t^2 - \nabla^2)H = 2\alpha H + e^2 \psi_0 (\phi^2 - A^2)$$

Quadratic term driving the Higgs
No phase fluctuations anymore.

Anderson-Higgs in Superconductors:

PHASE Transition & SYMMETRY Breaking

Spontaneous Symmetry Breaking (SSB)

- > In the **normal state**: Ground-state has the same U(1) symmetry than the potential
- > In the **SC state**: New ground-state has lower symmetry → spontaneous symmetry breaking

Let us consider an approach with **Amplitude and Phase fluctuations**:

Lagrangians of the condensate with an electromagnetic field tensor:

$$\begin{aligned} \mathcal{L}^{KG} &= (\partial_\mu \psi)^* (\partial^\mu \psi) - V(\psi) \\ &= (\partial_t \psi)^* (\partial_t \psi) - (\nabla \psi)^* (\nabla \psi) - V(\psi) \\ \mathcal{L}^{GP} &= \psi^* (\partial_t \psi) - (\nabla \psi^*) (\nabla \psi) - V(\psi) \end{aligned}$$

- Klein Gordon (Lorentz-invariant)
- (non-Lorentz invariant)
- Gross-Pitaevskii (BEC)

Perspective of the photons:

$$\mathcal{L} = -\frac{1}{2} ((\partial_\mu A_\nu)(\partial^\mu A^\nu) - (\partial_\mu A_\nu)(\partial^\nu A^\mu)) + e^2 \psi_0^2 A_\mu A^\mu + 2e^2 \psi_0 A_\mu A^\mu H$$

Euler-Lagrange

$$-(\partial_t^2 - \nabla^2) \mathbf{A} - \nabla(\nabla \cdot \mathbf{A}) - \partial_t \nabla \phi = 2e^2 \psi_0^2 \mathbf{A} + 4e^2 \psi_0 \mathbf{A} H$$

Static limit with vanishing H

$$\nabla^2 \mathbf{A} = 2e^2 \psi_0^2 \mathbf{A}$$

$$\nabla^2(\nabla \times \mathbf{A}) = 2e^2 \psi_0^2(\nabla \times \mathbf{A}) \rightarrow \nabla^2 \mathbf{B} = \frac{1}{\lambda^2} \mathbf{B} \quad \lambda = \sqrt{\frac{1}{2e^2 \psi_0^2}}$$

Meißner Effect determined by the inverse mass

Anderson-Higgs in Superconductors:

PHASE Transition & SYMMETRY Breaking

Spontaneous Symmetry Breaking (SSB)

- > In the **normal state**: Ground-state has the same U(1) symmetry than the potential
- > In the **SC state**: New ground-state has lower symmetry → spontaneous symmetry breaking

Let us consider an approach with **Amplitude and Phase fluctuations**:

Lagrangians of the condensate with an electromagnetic field tensor:

$$\begin{aligned} \mathcal{L}^{KG} &= (\partial_\mu \psi)^* (\partial^\mu \psi) - V(\psi) \\ &= (\partial_t \psi)^* (\partial_t \psi) - (\nabla \psi)^* (\nabla \psi) - V(\psi) \\ \mathcal{L}^{GP} &= \psi^* (\partial_t \psi) - (\nabla \psi^*) (\nabla \psi) - V(\psi) \end{aligned}$$

- Klein Gordon (Lorentz-invariant)
- (non-Lorentz invariant)
- Gross-Pitaevskii (BEC)

Adding the photons !!

$$\mathcal{L}^{GP} = (\psi_0 + H)(\partial_t H) + i(\psi_0^2 + 2\psi_0 H)((\partial_t \theta) + e\phi) - (\nabla H)^2 - \psi_0^2 (\nabla \theta)^2 + 2\alpha H^2$$

No local gauge transformation possible and accordingly no Anderson-Higgs possible.

Low-energy excitations:

- Case 1: Lorentz Invariant → Decoupled Amplitude and Phase Mode
- Case 2: Schrödinger Like → Only a Phase Mode
- Case 3: Charged Condensate Lorentz Invariant → Only a Higgs Mode

Point-by-Point Reply to the Reviewer's Comments:

Reviewer #1 (Remarks to the Author):

A: Regarding the Bardasis-Schrieffer mode, I don't agree with the authors' response.

B: Regarding whether a BEC of charged bosons has Meissner effect, I don't agree with the authors' response either. The presence of a charged superfluid with off diagonal long-range order leads directly to the Meissner effect. In the references (M.R. Schaffroth Phys. Rev. 100, 463 (1955) and Koh PRB 68, 144 502 (2003)) the authors provided, their conclusion is that the charged BEC does exhibit Meissner effect. Note that for charged bosons with interactions, the BEC means the "superfluid" phase. In discussion point (ii) on page 7 of Koh PRB 68, 144 502 (2003), it refers actually to the normal phase prior to the BEC phase, not the BEC case that the authors seem to take as. Koh clearly states that the BEC is a sufficient condition for Meissner effect in this paper.

However, we cannot argue endlessly in the peer reviewing process. Since these assertions in the reply (which I think are wrong) are not displayed in the revised manuscript, and because the manuscript has now weakened the corresponding claims, I can go with the current version of it. I think the manuscript is now publishable on nature communications.

We thank the reviewer for their continued engagement and for sharing their perspective on both the Bardasis-Schrieffer mode and the Meissner effect in a charged Bose-Einstein condensate (BEC). We appreciate the detailed clarification regarding the interpretation of the Koh PRB 68, 144502 (2003) paper. Regarding the Bardasis-Schrieffer mode, we note the referee's disagreement with our response. While we maintain our original view based on the context of our study, we respect the differing interpretation and agree that further discussion and experiments may extend beyond the scope of the current manuscript. We are grateful that, despite these remaining points of divergence, the referee finds the current version of the manuscript suitable for publication.